# Recent Advance in Co_3_O_4_ and Co_3_O_4_-Containing Electrode Materials for High-Performance Supercapacitors

**DOI:** 10.3390/molecules25020269

**Published:** 2020-01-09

**Authors:** Xuelei Wang, Anyu Hu, Chao Meng, Chun Wu, Shaobin Yang, Xiaodong Hong

**Affiliations:** 1College of Materials Science and Engineering, Liaoning Technical University, Fuxin 123000, China; wangxuelei-19@163.com (X.W.); huanyu990428@163.com (A.H.); mikko_mc@163.com (C.M.); chun_wu@126.com (C.W.); lgdysb@163.com (S.Y.); 2College of Mining, Liaoning Technical University, Fuxin 123000, China

**Keywords:** Co_3_O_4_, supercapacitor, composite, electrode material

## Abstract

Among the popular electrochemical energy storage devices, supercapacitors (SCs) have attracted much attention due to their long cycle life, fast charge and discharge, safety, and reliability. Transition metal oxides are one of the most widely used electrode materials in SCs because of the high specific capacitance. Among various transition metal oxides, Co_3_O_4_ and related composites are widely reported in SCs electrodes. In this review, we introduce the synthetic methods of Co_3_O_4_, including the hydrothermal/solvothermal method, sol–gel method, thermal decomposition, chemical precipitation, electrodeposition, chemical bath deposition, and the template method. The recent progress of Co_3_O_4_-containing electrode materials is summarized in detail, involving Co_3_O_4_/carbon, Co_3_O_4_/conducting polymer, and Co_3_O_4_/metal compound composites. Finally, the current challenges and outlook of Co_3_O_4_ and Co_3_O_4_-containing composites are put forward.

## 1. Introduction

Energy crisis and environmental pollution trigger the development of energy storage systems toward clean and renewable energies. Supercapacitors (SCs) act as a new type of energy storage devices between conventional capacitors and batteries [1]. The energy stored and released amount are related to the speed in energy storage devices [2]. In general, the greater the discharge power, the lower the energy that can be released [3]. The performance of SCs mainly involves specific capacitance (Cs), specific energy (E), specific power (P), resistance, cycling stability, and rate capacity. These are obtained by electrochemical cyclic voltammetry (CV), galvanostatic charge/discharge (GCD), and electrochemical impedance spectroscopy (EIS) [4]. The E of SCs is several hundred times higher than that of traditional capacitors, and the P is two orders of magnitude higher than that of the batteries. SCs make up the shortcomings of low specific power, poor charge, and discharge performance of large currents and low specific energy of conventional capacitors.

The above performances are closely related to the working principle of SCs. Based on the working principle, SCs involve electric double layer capacitors (EDLCs) and pseudocapacitors (PCs). EDLCs store charges using extremely thin double layer structure composed by the interface between electrodes and electrolytes, while PCs store energy by the reversible redox reactions on the electrode surface [5]. EDLC materials with high electronic conductivity and a large specific surface area have been used to achieve energy storage [6]. Due to the redox reaction, PCs can store more charges. The stored charges of EDLCs on the available surface are 0.17–0.20 electronics/atoms, while those of PCs are about 2.5 electronics/atoms. The stored energy by the PC system theoretically exceeds 10–100 times that of the same mass or volume of carbon-based EDLCs. However, the redox reaction of PCs induces the poor stability and low cycle life.

At present, carbon materials are the mainstream materials for the research and commercial application of EDLCs. They possess a high specific surface area, better electronic conductivity, and high chemical stability and they are abundant, low-cost, easy to process, and nontoxic. Common carbon materials include carbon nanotube (CNT), carbon fiber (CF), template carbon (TC), carbon aerogel (CA) and graphene, and so on [7,8,9,10,11]. Due to only physical adsorption being involved in charge storage, such EDLCs have an excellent charging and discharging stability, and the number of cycles even reaches several hundred thousand. Compared to EDLCs, PCs have a higher Cs and E [12], just for a large number of ions involved in the redox reaction [13]. Conducting polymers are typical PC electrode materials, while metal oxides are both EDLC and PC electrode materials [14,15,16]. Generally speaking, common conductive polymers include polyaniline, polypyrrole, polythiophene, and so on [17,18,19,20]. When charged and discharged, conductive polymers will undergo a rapid redox reaction to achieve the storage, release of charges, and produce a large capacity. The capacitance of conductive polymers is much higher than that of carbon materials. Conductive polymers have the advantages of good conductivity, low cost, abundant sources, and easy processing, but they have an unstable structure and poor cycling stability.

As a kind of PC electrode material, transition metal oxides mainly include RuO_2_, Co_3_O_4_, MnO_2_, and Fe_2_O_3_ [21,22,23,24]. As the surface of the atoms will be deposited at the underpotential, the generated charge in this process is just like chemical adsorption. This kind of material undergoes a reversible redox reaction of multiple electrons, so it stores many more charges [25]. Compared to other electrode materials, transition metal oxides have the higher Cs [26]. More and more researchers are studying the electrochemical performance of transition metal oxides. RuO_2_ is examined because of its high electrochemical properties [27]. However, its high price limits commercialization, while potential damage to the environment is also a barrier. Thus, other metal oxides have been developed as electrode materials, such as Co_3_O_4_, MnO_2_, Fe_2_O_3_, and so on [28]. Among them, Co_3_O_4_ electrode material has been widely studied because of the higher Cs, low price, and environmental friendliness. Furthermore, Co_3_O_4_ electrode material with special microstructures and morphology possesses an excellent electrochemical capacitive behavior [29]. However, Co_3_O_4_ electrode material has poor conductivity. In order to overcome the disadvantage of a single electrode material, the preparation of Co_3_O_4_-containing composites will achieve a superior combination performance [30]. Recently, graphene-based composites have been mostly studied [31] and acted as electrode materials for various energy storage systems [32]. For graphene, 2D porous graphene framework nanomaterials in SCs are a hotspot of recent research [33]. When combined with Co_3_O_4_ nanomaterial, they show excellent properties in asymmetric SCs [34]. Compared to a single component of Co_3_O_4_ and graphene, Co_3_O_4_/graphene composites exhibit a higher specific capacitance, specific energy, and specific power.

In this review, recent progress on Co_3_O_4_ and Co_3_O_4_-containing composites is summarized. Firstly, the synthetic methods and electrochemical performance of Co_3_O_4_ electrode materials are introduced, including the hydrothermal/solvothermal method, sol–gel method, thermal decomposition, chemical precipitation, electrodeposition, chemical bath deposition, and the template method. Then, Co_3_O_4_-containing electrode materials are summarized, including Co_3_O_4_/carbon, Co_3_O_4_/conducting polymer, and Co_3_O_4_/metal compound composites. The preparation and performance of various composite electrodes are discussed as given in Figure 1. Finally, the current challenges and outlook on Co_3_O_4_ and Co_3_O_4_-containing composites are put forward.

## 2. Synthesis and Performance of Co_3_O_4_

### 2.1. Hydrothermal or Solvothermal Method

The hydrothermal method is based on the change of solubility in the sealed heating stainless steel autoclave maintained at a certain temperature and pressure. Reaction temperature, time, pressure, and concentration of reactants will affect the morphology, size, and crystal type of the products [35]. Hydrothermal reaction has been widely used for the synthesis of Co_3_O_4_ nanomaterials. For example, one-dimensional (1D) nanorod-like Co_3_O_4_ was successfully synthesized via the hydrothermal method [36]. It had a high Cs of 655 F·g^−1^ at a current density of 0.5 A·g^−1^ and also exhibited a high capacitance retention and a better cycling stability. Further, two-dimensional (2D) ultrathin mesoporous Co_3_O_4_ nanosheets were generated on nickel foam (NF) substrate via a hydrothermal technique without any surfactant (Figure 2a) [37]. The Cs of Co_3_O_4_ nanosheets were as high as 610 F·g^−1^ at 1 A·g^−1^, and the cycling stability remained at about 94.5% after 3000 cycles at 4 A·g^−1^. Moreover, an ultrathin mesoporous Co_3_O_4_ nanosheet delivered an E of 136 Wh·kg^−1^ at the P of 0.75 kW·kg^−1^. In addition, a self-assembled microsphere material composed of Co_3_O_4_ nanosheets was synthesized via the hydrothermal method [38]. It showed a good cycling stability, and the capacitance remained about 101.7% after 1500 cycles at 1.0 A·g^−1^. The hydrothermal method was also used to synthesize 2D Co_3_O_4_ thin sheets consisting of three-dimensional (3D) interconnected nanoflake array [39]. The reaction temperature has a significant impact on the performance of SCs. Under 500 °C for 6 h, 2D Co_3_O_4_ thin sheets exhibited a Cs of 1500 F·g^−1^ at 1 A·g^−1^. Capacitance was retained at 99.3% after 2000 cycles at 5 A·g^−1^. Furthermore, as an electrode for an asymmetric SC, it exhibited an E of 15.4 Wh·kg^−1^ at a P of 0.8 kW·kg^−1^.

The difference between the solvothermal and hydrothermal method is that the organic phase is used as a solvent instead of or partially instead of water [40]. Under a high pressure, the physical properties of organic solvents (density, viscosity, and dispersion) will change greatly when compared to the atmospheric pressure condition, thus making it possible for some special chemical reactions to occur. Some solvents often participate in chemical reactions, such as ethanol, isopropanol, etc. At the same time, ethanol can also be used as a reducing agent in the system, and organic solvents can also be combined with reactants to change the state of the reactants. Thus, they affect the morphology, size, and other parameters of product [41]. This method has received extensive attention. For example, a solvothermal method was adopted to fabricate an ultrafine Co_3_O_4_ nanoparticle material [42]. These ultrafine Co_3_O_4_ nanoparticles exhibited a Cs of 523.0 F·g^−1^ at 0.5 A·g^−1^ and a good cycling stability, and Cs was retained at about 104.9% after 1500 cycles. Recently, Liu et al. [43] successfully fabricated homogeneous core–shell Co_3_O_4_ mesoporous nanospheres via the solvothermal method (Figure 2b). The core–shell structure effectively promoted electron and ion transmission and alleviated the strain of the Co_3_O_4_ electrode upon cycling. The Co_3_O_4_ electrode exhibited a high Cs of 837.7 F·g^−1^ at 1 A·g^−1^ and a remarkable rate capability. An asymmetric SC was constructed using Co_3_O_4_ nanowires as a positive and graphene aerogel as a negative electrode, which delivered an E of 35.8 W h·kg^−1^ at a P of 797.4 W·kg^−1^. Therefore, the solvothermal method provides a novel strategy to prepare Co_3_O_4_ electrode materials with different nanostructures.

**Figure 2 molecules-25-00269-f002:**
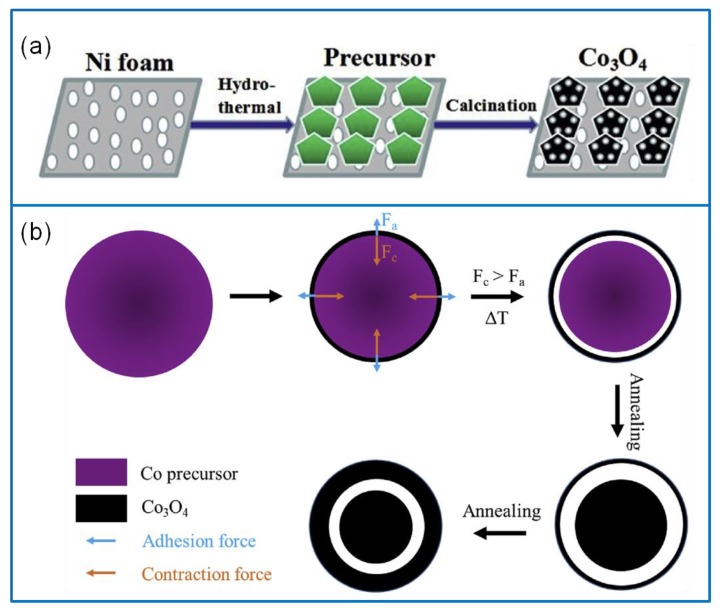
(**a**) Schematic illustration of preparing 2D ultrathin Co_3_O_4_ nanosheets via the hydrothermal method [37]. (**b**) Schematic illustration of synthesizing core–shell Co_3_O_4_ mesoporous nanospheres via the solvothermal method [43].

### 2.2. Sol–Gel Method

The sol–gel method is an important method for the synthesis of Co_3_O_4_ electrode materials. In a typical sol–gel method, an active precursor is evenly mixed in the liquid phase to form a transparent and stable sol through a hydrolysis and condensation process, and then aging to form a gel [44]. The sol–gel method has some advantages, such as low reaction temperature, ease of operation, miscellaneous and uniform molecular-level mixing, smaller size, and so on [45]. Co_3_O_4_ nanoparticles with no secondary phase were synthesized via the sol–gel method [46]. The Co_3_O_4_ nanoparticles delivered a Cs of 120 F·g^−1^ at 1 A·g^−1^ in a voltage window of 0 to 0.6 V. In addition, Lakehal et al. [47] prepared Co_3_O_4_ electrode materials by adopting the sol–gel-based dip-coating process on a glass substrate. Meanwhile, the effects of Ni-solution concentrations on the properties of Co_3_O_4_ samples were studied. The results showed that the resistance of obtained Co_3_O_4_ polycrystalline decreased in spinel-type, whereas the capacity increased with an increase in Ni doping levels. Peterson et al. [48] prepared Co_3_O_4_ with different morphologies via the sol–gel method and compared the performance of sphere-like, sponge-like, network-like, and plate-like Co_3_O_4_. The network-like sample exhibited a higher Cs of 708 F·g^−1^ at 5 mV·s^−1^ and a good rate capacity of 71.9% at 50 mV·s^−1^.

### 2.3. Thermal Decomposition

Thermal decomposition is widely used in the synthesis of Co_3_O_4_ electrode materials. In a typical thermal decomposition method, precursors are prepared by adding inorganic or organic salt solutions, and Co_3_O_4_ nanoparticles are synthesized via heating the precursors in air or another atmosphere [49]. The precursors gradually decompose into water, carbon dioxide, carbon and monoxide, etc. and leave a large number of Co_3_O_4_ pore structures. For instance, 1D porous Co_3_O_4_ nanowires were prepared via the thermal decomposition method with nitrilotriacetic acid (NA) as the chelating agent [50]. The Co_3_O_4_ nanowires exhibited a high Cs of 2815.7 F·g^−1^ at 1 A·g^−1^, an ideal rate capacity at 20 A·g^−1^, and an excellent coulombic efficiency (about 100%). Further, Ni foam supported Co_3_O_4_ nanoflakes were prepared via the thermal decomposition method [51]. Firstly, the nanosheet Co particles were deposited on Ni foam. Secondly, dendritic-like CoC_2_O_4_ nanowires were synthesized via in situ reaction. Finally, the dendritic-like CoC_2_O_4_ precursors were transformed into Co_3_O_4_ nanoflakes via the thermal decomposition method. The unique configuration showed a great advantage in terms of porosity and interlocking channels, because it effectively increased the surface contact with the electrolyte and ion/electron diffusion. The Co_3_O_4_ nanoflakes exhibited a high Cs of 576.8 F·g^−1^ at 1 A·g^−1^ and about 82% capacitance retention after 5000 cycles.

### 2.4. Chemical Precipitation

The chemical precipitation method can be classified into three categories: direct precipitation, coprecipitation, and homogeneous precipitation [52]. In a direct precipitation, reactants are directly added into a cobalt salt solution to produce precipitates, and then the product is washed, dried or calcined to obtain Co_3_O_4_ nanomaterials [53]. The method has the characteristics of simple operation and high product purity. In this regard, Wang et al. [54] prepared 3D-nanonet cobalt carbonate precursors via the direct precipitation method. The precursors were calcined in the air and converted into Co_3_O_4_ nanoparticles with original frame structures (Figure 3a), which displayed a Cs of 739 F·g^−1^ at 1 A·g^−1^ and a capacitance retention of 90.2% after 1000 cycles at 5 A·g^−1^. The coprecipitation method refers to the deposition of multiple metal ions [55]. Therefore, it is rarely used for the synthesis of a single Co_3_O_4_ nanomaterial. For the homogeneous precipitation method, metal ions are slowly released to the solution through a chemical reaction. Thus, homogeneous precipitation reactions do not occur immediately between precipitated ions. This overcomes the disadvantage of local inhomogeneity caused by the direct introduction of precipitates. The Co_3_O_4_ nanomaterials synthesized via homogeneous precipitation have a uniform size distribution and high purity. For example, an ultrathin Co_3_O_4_ material with high porosity was synthesized via a homogeneous precipitation method [56]. The structure was composed of well-arranged 2D rectangular thin sheets with a high specific surface area, pore volume, and uniform aperture distribution (Figure 3b). The Co_3_O_4_ thin sheets showed a high Cs of 548 F·g^−1^ at 8 A·g^−1^, and a capacitance retention of 98.5% after 2000 cycles at 16 A·g^−1^. They also exhibited an excellent cyclic stability, structural stability, and electrochemical stability.

### 2.5. Electrodeposition

With this method, nanomaterials are deposited on the electrode surface via an electrochemical reduction reaction [57]. The active materials can be evenly distributed on the electrodes through this method. In this technique, the morphology, size, and composition of synthesized nanomaterials are controllable. For example, Co_3_O_4_ nanoplates were prepared via the electrodeposition method [58]. Initially, Co(OH)_2_ nanoplates were synthesized through an additive-free electrodeposition route, and then the Co_3_O_4_ nanoplates were obtained via calcination. The Co_3_O_4_ nanoplates exhibited a Cs of 485 F·g^−1^ and a capacitance retention of 84.1% after 3000 cycles at 5 A·g^−1^. Further, a Co_3_O_4_ nanosheet was prepared through a one-step cathodic electrophoretic deposition of polyethylenimine in aqueous solutions [59]. The electrode showed a Cs of 233.6 F·g^−1^ at 0.5 A·g^−1^ and a capacitance retention 93.5% after 2000 cycles. Meanwhile, other Co_3_O_4_ nanosheets were fabricated on Ni foam via the electrodeposition of a Co(OH)_2_ precursor following annealing [60]. The Co_3_O_4_ electrode presented a super high Cs of 6469 F·g^−1^ at 5 mA·cm^−2^, which exceeded its theoretical value. Impressively, it maintained a Cs of 4127 F·g^−1^ at 15 mA·cm^−2^ and a capacitance retention of 81.6% after 2000 cycles.

### 2.6. Chemical Bath Deposition

Chemical bath deposition (CBD) is a chemical reduction process. A suitable reducing agent is used to reduce the metal ions in the plating liquid, and the metal ions are deposited on the matrix of the matrix. CBD directly grows quantum dots on substrate, and it is a relatively slow chemical reaction process. Compared to other preparation methods, the CBD method has obvious advantages in its controllability, uniformity, and low cost. Moreover, different substrates and solutions can be used to prepare Co_3_O_4_ electrode materials [61]. For example, Co_3_O_4_ nanowire was synthesized on stainless steel (SS) substrate via the CBD method [62], which possessed a high specific surface area of 66.33 m^2^·g^−1^. It exhibited a high Cs of 850 F·g^−1^ at 5 mV·s^−1^ and a cycling stability of about 86% after 5000 cycles. The symmetric SC device was fabricated using a Co_3_O_4_ nanowire, which showed a Cs of 127 F·g^−1^, an E of 24.18 Wh·kg^−1^, and a cycling stability of 85% after 3000 cycles. In addition, Co_3_O_4_ nanorod arrays were fabricated via the facile CBD method [63]. The size of the nanorod was about 450 nm. As an electrode, the Co_3_O_4_ nanorod arrays delivered a Cs of 387.3 F·g^−1^ at 1 A·g^−1^ and a cyclic stability of 88% after 1000 cycles.

### 2.7. Template Method

The template method usually adopts a template with a certain morphology and nanostructure. The products are formed on the template, then the template is removed to obtain nanomaterials with the same morphology and size [64]. The most used templates are surfactants, biomolecules, polymers, and a metal–organic framework (MOF) [65]. For example, a 2D hierarchical Co-based MOF of UPC-9 was used to synthesize ultrathin and rich macroporous Co_3_O_4_ nanosheets [66]. It exhibited a remarkable Cs of 1121 F·g^−1^ at 1 A·g^−1^ and a rate capability of 77.9% at 25 A·g^−1^. This is because the organic ligand of UPC-9 prevents the agglomeration of Co_3_O_4_ caused by calcination. In addition, another ultrathin Co_3_O_4_ nanosheet was constructed by the zeolitic imidazolate framework-67 (Figure 4a) [67]. The Co_3_O_4_ electrode showed a large Cs of 1216.4 F·g^−1^ at 1 A·g^−1^ and a higher rate capability of 76.1% at 20 A·g^−1^. The template method can also be used to prepare other morphological Co_3_O_4_ electrode materials. The hollow spherical Co-BTB-I and flower-like Co-BTB-II precursors were obtained via the template method with/without hexadecyl trimethyl ammonium bromide (CTAB) (Figure 4b) [68]. After annealing, the Co_3_O_4_ nanoparticles maintained the template morphologies. The Co-BTB-I electrode exhibited a Cs of 342.1 F·g^−1^ at 0.5 A·g^−1^. The template method has many advantages; however, it needs more processing steps and has certain requirements on the template.

In addition to the above methods, there are other methods to synthesize Co_3_O_4_, such as spray pyrolysis [69], chemical vapor deposition (CVD) [70], the electrospinning technique [71], galvanic displacement [72], laser ablation [73], the in situ self-organization method [74], and so on. To sum up, there are many synthesis methods for preparing Co_3_O_4_ electrode materials, which are mainly divided into three categories: the gas phase method, solid phase method, and liquid phase method. The Co_3_O_4_ electrode materials synthesized via the gas phase method have the advantages of small size, good dispersion, etc. However, they need tough experimental equipment and conditions. The materials prepared via the solid phase method have the advantages of no agglomeration and good filling but have the disadvantages of high energy consumption, low efficiency, and many more impurities. The liquid phase method is one of the most commonly used methods to prepare Co_3_O_4_ electrode materials. The liquid phase method has the advantages of simple operation, mild conditions, high purity, and good dispersion. However, the development of Co_3_O_4_ electrode materials is not limited to the laboratory stage; we should consider the industrialization and commercialization of Co_3_O_4_. The preparation of high-performance Co_3_O_4_ electrode materials requires tough synthesis conditions, such as, high temperature and pressure, with a small-scale output, which affect the rapid development of industrialization and commercialization to some extent. Therefore, developing a low-cost, rapid, and simple synthesis method is an important task for improving the performance of Co_3_O_4_.

### 2.8. Performance Statistics of Different Co_3_O_4_ Materials

The relationship between the electrochemical performance, synthetic method, and the microstructures of Co_3_O_4_ is summarized in Table 1. Among them, Co_3_O_4_ nanoparticles synthesized via the hydrothermal method have a good dispersion, high purity, and high crystallinity. Compared to the hydrothermal method, the solvothermal method tends to synthesize smaller particles. For the sol–gel method, when the gel is formed, the reactants are likely to be evenly mixed at the molecular level, and only low reaction temperature is required. The thermal decomposition method and chemical precipitation method are simple, adaptable, and controllable. The electrodeposition method is generally used to prepare thin film electrodes. Unlike electrochemical deposition, CBD does not require a rectifier power supply and anode. Moreover, the synthesized electrode materials are an almost nanorod structure. As for the template method, as-prepared electrode materials often have a high Cs. Among those Co_3_O_4_ materials, Co_3_O_4_ nanosheets prepared via the electrodeposition method displayed an ultrahigh Cs of 6469 F·g^−1^ with cycling stability of 81.6% after 2000 cycles [60]. The Co_3_O_4_ nanospheres synthesized via the solvothermal method exhibited a Cs of 837.7 F·g^−1^ at 1 A·g^−1^ and an excellent rate capacity of 93.6% at 10 A·g^−1^ [43]. Another Co_3_O_4_ nanosphere showed a cyclic stability greater than 100% after 4000 cycles via the CVD method [70].

## 3. Advance of Co_3_O_4_-Containing Composites

### 3.1. Co_3_O_4_/Carbon Composites

#### 3.1.1. Co_3_O_4_/Carbon Nanotube (CNT) Composites

CNT usually divides into a single-walled carbon nanotube (SWCNT) and multiwalled carbon nanotube (MWCNT). They are seamless hollow tubes that are curled by single-layer or multilayer graphite sheets. They have a unique pore structure, high electrical conductivity, excellent mechanical properties, and prominent surface area utilization and thermal stability, so they are extremely suitable for supercapacitor electrode materials [75]. However, CNT also has some disadvantages, such as a low specific surface area and expensive price. In order to improve its performance, CNT is usually used as carriers to deposit PC materials to prepare composites [76]. Therefore, Co_3_O_4_/CNT composites have been extensively studied [77]. SWCNT thin film was prepared via a vacuum filtration and continuously stamping method, and then Co_3_O_4_/SWCNT composite was obtained via the electrodeposition method. It exhibited a Cs of 70.5 mF·cm^−2^ at 1 mV·s^−1^ and a capacitance retention of 80% after 3000 cycles [78]. Further, Co–Co_3_O_4_/N–SWCNT core–shell composite was synthesized via simple pyrolysis of cobalt acetate and melamine–formaldehyde resin in N_2_ atmosphere [79]. Due to the close contact between the Co–Co_3_O_4_ nanoparticle cores and the N–SWCNT shells, Co–Co_3_O_4_@N-SWCNT showed a high Cs of 823.4 F·g^−1^ at 1 A·g^−1^ and a capacitance retention of 93.6% over 10,000 cycles. Moreover, an asymmetric device of Co–Co_3_O_4_/ CNT–NC//reduced graphene oxide SC exhibited a high E of 46.7 Wh·kg^−1^ at a P of 1601.1 W·kg^−1^. In addition, a nitrogen-doped MWCNT/Co_3_O_4_ composite was prepared in the presence of urea and aqueous ammonia via the thermal decomposition method [80]. The Co_3_O_4_ nanoparticles were densely dispersed on the N-MWCNT surface. The Co_3_O_4_/N-MWCNT electrode material exhibited a high Cs of 406 F·g^−1^ at 2 A·g^−1^ and an excellent capacitance retention of 93% after 10,000 cycles.

#### 3.1.2. Co_3_O_4_/Carbon Fiber (CF) Composites

CF has a moderately controllable aperture distribution, and its pore structure is suitable for the transfer of electrolyte ions. Thus, it is often used as a carrier of electrode materials [81]. Recently, there has been extensive focus on the development of composite nanofiber materials for enhancing the Cs, E, and P of SCs [82]. For example, a Co_3_O_4_/CF composite was prepared via the electrospinning method followed by heat treatment [83]. Onion-shaped graphitic layers were formed around the Co_3_O_4_ nanoparticles, which improved the electrical conductivity of the electrode and prevented the Co_3_O_4_ nanoparticles from falling from the CF matrix. The Co_3_O_4_/CF composite delivered a Cs of 586 F·g^−1^ at 1 A·g^−1^ and a capacitance retention of 74% after 2000 cycles at 2 A·g^−1^. Further, 3D Co_3_O_4_ nanowire arrays were grown on CF via the CVD method (Figure 5a) [84]. The Co_3_O_4_/CF electrode material exhibited a Cs of 734.25 F·cm^−3^ (2210 mF·cm^−2^) at 1.0 A·cm^−3^. Moreover, all-solid-state fiber-shaped asymmetric SC composed of vanadium nitride nanowires/CF//Co_3_O_4_/CF exhibited a window of 1.6 V and an E of 13.2 mWh·cm^−3^ at 1.0 A·cm^−3^. Recently, Co_3_O_4_/biomass-derived carbon fiber (BCF) with a hierarchical structure was fabricated as shown in Figure 5b [85]. Hollow porous BCF was treated as the sandwich layer, and Co_3_O_4_ nanoparticles were used for inner and outer cladding. The porous BCF not only provided an ideal electron transfer path to surmount the limit of high resistance of the Co_3_O_4_ electrode but also served as a backbone, which facilitated the loading of more Co_3_O_4_ particles to promote the redox reaction. The Co_3_O_4_/BCF composite delivered a Cs of 892.5 F·g^−1^ at 0.5 A·g^−1^ and a capacitance retention of 88% over 6000 cycles.

Carbon cloth (CC) is made of CF. It has the advantages of low cost, good conductivity, light quality, and excellent flexibility [86]. CC can be applied to support cobalt oxide electrode material. The resulting composite can be used directly without secondary processing. This effectively reduces the resistance of the electrode and makes the electrode response good at a high current density. Therefore, CC is widely used in flexible SCs. For example, cobalt oxide was loaded on CC via the cathodic potentiodynamic procedure [87]. The supercapacitor capacity was related to the content of the cobalt oxide. The composite with 8 wt% cobalt oxide on CC exhibited a Cs of 227 mAh·g^−1^ at 1 mA·cm^−2^ and a capacitance retention of 82% after 5000 cycles.

#### 3.1.3. Co_3_O_4_/Template Carbon (TC) Composites

At present, TC is commonly used carbon material for SCs [88]. It is a porous carbon material with a uniform and concentrated pore size distribution [89]. There are two ways to prepare TC. The first one is also called active carbon (AC). It is usually derived from physically and chemically activated wood, petroleum coke, phenolic resin, and sucrose. Another is used carbon precursors to infiltrate into template pores, and then the template is removed to obtain porous carbon, which is opposite to the template [90]. Based on the first method, a Co_3_O_4_/AC composite was synthesized via the microwave-assisted deposition-precipitation method [91]. When the loading of Co_3_O_4_ is 16.4 wt%, the average size of Co_3_O_4_ nanoparticles is 7 nm (Figure 6a). This Co_3_O_4_/AC electrode material presented a Cs of 491 F·g^−1^ at 0.1 A·g^−1^ and a capacitance retention of 89% over 5000 cycles at 5 A·g^−1^ (Figure 6b). Further, a hybrid Co_3_O_4_/AC electrode material was synthesized from alginate and cobalt salt via a simple self-crosslinking and pyrolysis method (Figure 6c) [92]. The Co_3_O_4_/AC hybrid material prepared by alginate had the advantages of low cost, a simple manufacturing process, and excellent electrochemical performance. In addition, 3D carbon foam is excellent AC material, and it is prepared via the carbonization of phenolic foam [93]. The unique structure increases the stored amount of charges. For example, a cobalt oxide/carbon foam composite was prepared using the solvothermal method [94], which had a wide pore size distribution and a large specific surface area. It exhibited a high Cs of 115 mAh·g^−1^ at 0.5 A·g^−1^, a rate capability of 73 mAh·g^−1^ at 10 A·g^−1^, and a capacitance retention of 86% at 1 A·g^−1^ after 6000 cycles.

Using the carbon precursors, a 3D hierarchical carbon skeleton/Co_3_O_4_ composite was prepared [95]. The 3D hierarchical carbon skeleton was fabricated on the silica nanosphere templates via thermal vapor deposition. Phosphoric acid was employed to control its porosity and surface area. The TC electrode exhibited a Cs of 134 F·g^−1^ at 10 mV·s^−1^, while the resulting composite had an enhanced Cs of 456 F·g^−1^ at 1 A·g^−1^. Further, a zeolitic imidazolate framework-67 precursor was used to synthesize the Co_3_O_4_/TC composite, which exhibited a high Cs of 885 F·g^−1^ at 2.5 A·g^−1^ and a capacitance retention of 94% over 10,000 cycles [96]. Recently, a mesoporous carbon (CMK-3) was prepared via the hard template of silica SBA-15, and the silica template was removed via HF [97]. The Co_3_O_4_/CMK-3 composite was grown on NF by a hydrothermal and annealing process. It exhibited a high Cs of 1131.3 F·g^−1^ at 0.5 A·g^−1^ and a capacitance retention of 91% after 3000 cycles, while the bare Co_3_O_4_ film delivered a Cs of 727 F·g^−1^ and 82% capacitance retention.

#### 3.1.4. Co_3_O_4_/Carbon Aerogel (CA) Composites

CA is a kind of nanoporous amorphous material; it has a unique three-dimensional network structure, with light weight, a large specific surface area, good conductivity, and rich dielectric electrochemical stability [98]. For hierarchical porous CA, macropores can promote electron transmission to increase the specific power, while mesopores and micropores are responsible for providing large specific surface areas to increase the E. In the field of Co_3_O_4_/CA composites, a hybrid Co_3_O_4_/CA electrode was synthesized via the in situ growth method [99]. The Co_3_O_4_/CA electrode material exhibited a Cs of 350 F·g^−1^ at 1 A·g^−1^ and an E of 23.82 kW·kg^−1^ at a P of 95.96 W·kg^−1^. In a symmetrical SC device, it could be cycled reversibly in a voltage window of 0.0 to 1 V at 1 A·g^−1^ and exhibit a capacity retention of 210% over 6000 cycles. Furthermore, CA can be modified by other materials. For example, a novel high-performance Co_3_O_4_/nitrogen-doped carbon aerogel (NCA) electrode material was prepared via the in situ coating method followed by the freeze-drying process [100]. The Co_3_O_4_/NCA composite exhibited a Cs of 616 F·g^−1^ at 1 A·g^−1^ and an excellent rate capability of 445 F·g^−1^ at 20 A·g^−1^. The assembled asymmetric Co_3_O_4_/NCA//NCA SC device could be cycled in a range of 1.5 V with an E of 33.43 Wh·kg^−1^ at a P of 375 W·kg^−1^. Further, a hierarchical porous carbonaceous aerogel (HPCA) was fabricated using renewable seaweed aerogel [101]. It possessed a hierarchical micro/meso/macroporous structure and a high specific surface area of 2200 m^2^·g^−1^. The HPCA exhibited a Cs of 260.6 F·g^−1^ at 1 A·g^−1^ and a capacitance retention of 91.7% over 10,000 cycles at 10 A·g^−1^. When the HPCA was used to grow Co_3_O_4_ nanowires, the Co_3_O_4_/HPCA exhibited a high Cs of 1167.6 F·g^−1^ at 1 A·g^−1^.

#### 3.1.5. Co_3_O_4_/Graphene Composites

Graphene has a high specific surface area and excellent mechanical properties and electrochemical properties. It shows excellent properties and has great potential in the application of SC [102]. Compared with other materials, graphene has a high conductivity, high carrier mobility, and excellent mechanical strength. Graphene and its derivatives can be directly used as electrode materials for SCs [103]. However, graphene undergoes irreversible agglomeration and restacking during the combination. Discovering how to avoid the agglomeration and stacking problem of graphene is the key to preparing high-performance graphene electrodes [104]. The combination of graphene with metal oxides is an effective way to solve these problems [105]. For example, a hybrid Co_3_O_4_ nanosheet on the graphene@NF electrode was designed via the in situ synthesis method (Figure 7a) [106]. Due to the different surface diffusion coefficients of Co_3_O_4_ nanoparticles, the thickness of Co_3_O_4_ nanosheets can be reduced from 70 to 13 nm with an increase of graphene sheets. The Co_3_O_4_/graphene@Ni hybrid composite with a 13 nm thick nanosheet exhibited a high Cs of 1.75 F·cm^−2^ at 1 mA·cm^−2^ and a capacitance increase of 12.2% after 5000 cycles at 10 mA·cm^−2^. As a typical carbon material, reduced graphene oxide (rGO) has a high dielectric constant and chemical stability. It has been used as an encapsulator to improve the electrochemical properties of SCs. A Co_3_O_4_/rGO electrode composite was fabricated via the temperate coprecipitation method. GO was used as a substrate, and a ZIF-67 rhombic dodecahedron was used as the template (Figure 7b) [107]. The Co_3_O_4_/rGO composite electrode exhibited a Cs of 546 F·g^−1^ at 0.5 A·g^−1^, a rate capability of 90.8% at 5 A·g^−1^, and a capacitance retention of 90% over 10,000 cycles at 5 A·g^−1^.

Improving E under high P is an urgent problem in the design of SCs. Graphene and its derivatives have done well in this field. A Co_3_O_4_/graphene composite was hydrothermally deposited on an NF substrate [34]. The cross-linked porous Co_3_O_4_ nanofiber array on the graphene sheet showed a good boost of performance. The pseudocapacitive Co_3_O_4_/graphene electrode offered a high Cs of 1935 F·g^−1^ at 5 A·g^−1^, a rate capability retention of 68% within 0.5–50 A·g^−1^, and a capacitance decline of 17% within 2000 cycles. The assembled asymmetric Co_3_O_4_/grapheme//AC device exhibited an E of 50.3 Wh·kg^−1^ at a P of 786 W·kg^−1^. Additionally, a Co_3_O_4_/nitrogen-doped graphene (G > N) composite was synthesized via the solvothermal method [108]. The flower-like Co_3_O_4_ was loaded on the G > N nanosheets, which were modified with methoxypolyethylene glycol (mPEG) (Figure 7c). The Co_3_O_4_/G > N electrode materials exhibited a high Cs of 1625.6 F·g^−1^ at 0.5 A·g^−1^. Particularly, the assembled asymmetric Co_3_O_4_-G > N//rGO-CNT > N aerogel device exhibited an E of 34.4 Wh·kg^−1^ at a P of 400 kW·kg^−1^.

### 3.2. Co_3_O_4_/Conductive Polymer Composites

Conductive polymers mainly include polyaniline (PANI), polypyrrole (PPy), polythiophene (PTh), and their derivatives. They have advantages of high stability, ease of process, and excellent electrochemical properties. Thus, they are often used as electrode materials [109]. However, conductive polymers lose their conductivity due to electrode damage caused by the embedding and deactivation of anti-ions during circular charging and discharging. This reaction characteristic limits the development of conductive polymers in SCs. In order to further improve the electrochemical performance of conductive polymers, researchers have generally attempted to prepare Co_3_O_4_/conducting polymer composites [110].

#### 3.2.1. Co_3_O_4_/PANI Composites

PANI has a good conductivity, high specific capacitance, and high oxidation activity [111]. It is the most widely used conductive polymer in combination with Co_3_O_4_ composites [112]. A core–shell structured Co_3_O_4_/PANI electrode material with excellent electrical conductivity and ion diffusion behavior was synthesized via the in situ polymerization method (Figure 8a) [113]. It exhibited a high Cs of 1184 F·g^−1^ at 1.25 A·g^−1^ and a capacitance retention of 84.9% after 1000 cycles. Furthermore, a hierarchical hollow Co_3_O_4_/PANI electrode material was also prepared via the in situ polymerization route (Figure 8b) [114]. Due to the excellent conductivity of PANI and hollow nanocage structure, the electron transport rate and specific surface area of the Co_3_O_4_/PANI electrode material were improved. The contact resistance and charge-transfer resistance of Co_3_O_4_/PANI electrode material were significantly lower than those of the pristine Co_3_O_4_. The Co_3_O_4_/PANI electrode material exhibited a high Cs of 1301 F·g^−1^ at 1 A·g^−1^ and a cycling retention of 90% after 2000 cycles. The assembled Co_3_O_4_/PANI//AC device delivered an E of 41.5 Wh·kg^−1^ at 0.8 kW·kg^−1^, and a P of 15.9 kW·kg^−1^ at 18.4 Wh·kg^−1^. In addition, a Co_3_O_4_/PANI layered thin film was successfully prepared on SS substrate via the electrodeposition technique in 0.5 M Na_2_SO_4_ electrolyte [115]. Compared to the original Co_3_O_4_ electrode material, it exhibited nearly a 50% increase in the Cs and a significant enhancement of E and P. Particularly, the layered composite electrode showed a capacitance retention of 100% after 500 cycles.

#### 3.2.2. Co_3_O_4_/PPy Composites

PPy is a new conductive polymer; it has the advantages of good environmental stability, high conductivity, easy synthesis, good extensibility, and excellent electrical characteristic [116]. A lot of works have been reported on Co_3_O_4_/PPy composites [117]. A hierarchical Co_3_O_4_/PPy nanowire composite was successfully prepared via in situ chemical polymerization with a template-free hydrothermal route [118]. The Co_3_O_4_ nanowires were evenly generated on an ultrathin layer of amorphous PPy (Figure 9a). The Co_3_O_4_/PPy composite had highly electronic conductivity and electroactivity, which could significantly increase active sites and reduce charge transfer resistance. On the basis of these merits, the Co_3_O_4_/PPy electrode material exhibited a high Cs of 2122 F·g^−1^ at 5 mA·cm^−2^ and a capacitance retention of 77.8% after 5000 cycles at 25 mA·cm^−2^. In addition, a unique Co_3_O_4_/PPy core–shell nanorod electrode material was prepared via the hydrothermal and electrochemical deposition method [119]. The SC device was fabricated using Co_3_O_4_/PPy as the positive electrode and active CF as the negative electrode. Due to a wide voltage window of 1.5 V, the device exhibited a high areal capacitance of 1.02 F·cm^−2^ and a perfect cycling stability of 98% after 5000 cycles at 50 mA·cm^−2^. Recently, a flexible, high-performance, and tailored nanorod solid-state SC was assembled using a Co_3_O_4_/PPy electrode, a porous carbon electrode, and the PVA electrolyte (Figure 9b) [120]. Benefiting from the Co_3_O_4_ nanorods and conductive PPy layer, the Co_3_O_4_/PPy electrode material exhibited a high areal capacitance of 6.67 F·cm^−2^ at 2 mA·cm^−2^. Moreover, the solid-state SC can be tailored into multiple units and various shapes.

#### 3.2.3. Co_3_O_4_/Other Conductive Polymer Composites

The poly(3,4-ethylenedioxythiophene) (PEDOT) has attracted wide attention in recent years. It has a high conductivity, rapid redox, good stability, good film formation, and a high specific surface area. The Co_3_O_4_/PTh composite was synthesized via in situ chemical oxidative polymerization [121]. In addition, the Co_3_O_4_/PEDOP nanorod was fabricated on flexible carbon–fabric substrates [122], which delivered a Cs of 407 F·g^−1^ at 1 A·g^−1^, a retention of 78% over 5000 cycles, and an E of 29.9 Wh·kg^−1^ at a P of 0.15 kW·kg^−1^. Further, polyindole (Pind) is a common conductive polymer material. A Co_3_O_4_/Pind electrode material was prepared via the in situ cathodic electrodeposition method [123]. The Co_3_O_4_/Pind exhibited a high Cs of 1805 F·g^−1^ at 2 A·g^−1^ and an excellent rate capability of 1625 F·g^−1^ at 25 A·g^−1^.

### 3.3. Co_3_O_4_/Metal Compound Composites

#### 3.3.1. Co_3_O_4_/Metal Oxide Composites

Metal oxide electrode materials are considered to hybridize with Co_3_O_4_ [124]. Thus, more and more metal oxides composites containing Co_3_O_4_ have been reported, such as RuO_2_, NiO, Fe_2_O_3_, ZnO, CuO, MnO_2_, SnO_2_, and CoO [125,126,127,128,129,130]. For example, a fish thorn-like nonstructural Co_3_O_4_/NiO electrode material was successfully synthesized [131] with a core–shell-like structure of NiO nanosheet arrays (NiO/Co_3_O_4_/NiO) (Figure 10a). It exhibited a Cs of 313.9 μAh·cm^−2^ at 4 mA·cm^−2^. The assembled NiO/Co_3_O_4_/NiO//AC device showed a Cs of 623.5 mF·cm^−2^ at 2 mA·cm^−2^, an E of 216.1 μWh·cm^−2^ at a P of 27.7 mW·cm^−2^, and a prominent capacity retention of 126% over 5000 cycles. Furthermore, a hierarchical flower-like Co_3_O_4_/ZnO nanobundle electrode was synthesized via the hydrothermal method [132] (Figure 10b). It exhibited a remarkable Cs of 1983 F·g^−1^ at 2 A·g^−1^ and a capacitance retention of 84.5% at 10 A·g^−1^ over 5000 cycles. The Co_3_O_4_/ZnO//graphene asymmetric Sc device showed an ultrahigh E of 70.4 Wh·kg^−1^ at a P of 779.8 W·kg^−1^.

Further, a Co_3_O_4_/CuO electrode with a 1D nanowire morphology was prepared via the electrospinning technique [133]. The Cs values were 712, 1104, and 1242 F·g^−1^ for CuO, Co_3_O_4_, and Co_3_O_4_/CuO electrode materials at 2 mV·s^−1^, respectively. Moreover, the Co_3_O_4_/CuO//AC device exhibited a high E of 44 Wh·kg^−1^ at a P of 14 kW·kg^−1^ and a capacitive retention of 99% after 2000 cycles at 5 A·g^−1^. These were related to the voltage window of 1.6 V. For Co_3_O_4_/CoO electrode material, a Co_3_O_4_/CoO core–shell nanocrystal with a unique mesoporous microsphere structure was prepared via the solvothermal method following annealing treatment [134]. It exhibited an ultrahigh Cs of 3377.8 F·g^−1^ at 2 A·g^−1^. Furthermore, the Co_3_O_4_/CoO//graphene device delivered a high E of 44.06 Wh·kg^−1^ at a P of 800 W·kg^−1^. As for MnO_2_, the Co_3_O_4_/MnO_2_ core–shell arrays were grown on NF for SC via the two-step hydrothermal method [135]. It showed a high Cs of 1920 F·g^−1^ at 1 A·g^−1^ and a capacitance retention of 95.2% after 3000 cycles.

In addition to the composites of metal oxide and Co_3_O_4_, doped Co_3_O_4_ has been reported. Metal doping is that the metal ions enter the interior of the Co_3_O_4_ crystal and substitute Co atom without changing the crystal structure [136]. Metal doping mainly involves Mn [137], Fe [138], and Cd [139]. An Mn-doped Co_3_O_4_ mesoporous nanoneedle was prepared via the one-step hydrothermal method followed by annealing the precursor on NF [140]. The Mn-doping Co_3_O_4_ electrode material showed an excellent retention of 104% after 10,000 cycles at 6 A·g^−1^. Further, Fe-doping Co_3_O_4_ electrode material was hydrothermally synthesized [138], which exhibited a high Cs 1997 F·g^−1^ at 1 A·g^−1^ and an excellent rate capability of 1757 F·g^−1^ at 20 A·g^−1^. When tested in the voltage window of 0–1.8 V, the SC device delivered an E of 270.3 Wh·kg^−1^ at 224.2 Wh·kg^−1^ and a retention of 91.8% over 10,000 cycles at 10 A·g^−1^.

#### 3.3.2. Co_3_O_4_/Ternary Metal Oxide Composites

Compared to common metal oxide, ternary metal oxide has a better conductivity and higher redox activity in water electrolytes [141]. These mainly include NiCo_2_O_4_ NiMoO_4_, CoWO_4_, ZnCo_2_O_4_, ZnFe_2_O_4_, CoFe_2_O_4_, CoMn_2_O_4_ MnCo_2_O_4_, etc. [142,143,144]. These ternary metal oxides are often used as SCs electrode materials combining with Co_3_O_4_ [145,146,147,148,149]. For example, a hierarchical core–shell Co_3_O_4_/NiCo_2_O_4_ electrode material was successfully fabricated on NF [150]. It exhibited an excellent Cs of 1330 F·g^−1^ at 3 mA·cm^−2^ and a capacity retention of about 100.7% over 5000 cycles. Further, tube-like yolk–shell Co_3_O_4_/NiMoO_4_ electrode material was synthesized via the two-step method [151], which consisted of ultrathin porous shell of NiMoO_4_ and Co_3_O_4_ fiber (Figure 11a). Benefitting from the unique structure and the synergistic effect of NiMoO_4_ nanosheets and Co_3_O_4_ fibers, the Co_3_O_4_/NiMoO_4_ electrode exhibited a Cs of 913.25 F·g^−1^ at 10 A·g^−1^, a capacitance retention of 88% at the current density ranging from 0.5 to 20 A·g^−1^, and a cycling stability of 89.9% over 3000 cycles at 20 A·g^−1^. As for CoWO_4_, 3D CoWO_4_/Co_3_O_4_ heterostructure electrode material was synthesized via the microwave hydrothermal method [152] (Figure 11b). It showed a high capacitance of 4.665 F·cm^−2^ at 2.7 mA·cm^−2^ and a rate performance of 70.83% at 27 mA·cm^−2^. Moreover, the device was composed of the porous carbon as anode and CoWO_4_/Co_3_O_4_ as cathode, which exhibited a high E of 45.6 Wh·kg^−1^ at a P of 750 W·kg^−1^, and a capacitance retention of 90.5% after 5000 cycles.

In addition, a hollow starfish-shaped porous Co_3_O_4_/ZnFe_2_O_4_ electrode material was reported [153]. It exhibited a Cs of 326.7 F·g^−1^ at 1 A·g^−1^, a capacitance retention of 51.8% at 10 A·g^−1^, a cycling capacitance retention of 80.7% after 1000 cycles, and a high E of 82.5 Wh·kg^−1^ at a P of 675 W·kg^−1^. The performance of the Co_3_O_4_/ZnFe_2_O_4_ composite was much higher than that of the single component. Recently, Sapna et al. [154] synthesized a heterostructured Co_3_O_4_/CoFe_2_O_4_ electrode material using the hydrothermal method. It exhibited a high Cs of 761.1 F·g^−1^ at 10 mV·s^−1^ and a capacitance retention of 92.2% after 1000 cycles. In another work, a 3D hierarchical shell/core nanosheet/nanoneedle Co_3_O_4_/CoMn_2_O_4_ electrode material was synthesized on NF [155]. It exhibited a high Cs of 1627 F·g^−1^ at 1 A·g^−1^, a rate capability of 84.6% at 10 A·g^−1^, and a capacitance retention of 87.6% over 3000 cycles at 4 A·g^−1^. The assembled asymmetric Co_3_O_4_/CoMn_2_O_4_//AC SC device showed a Cs of 125.8 F·g^−1^ at 1 A·g^−1^, a retention of 89.2% after 5000 cycles at 2 A·g^−1^, and a high E of 44.8 W h·kg^−1^ at a P of 800.5 W·kg^−1^. In particular, the MnCo_2_O_4_/Co_3_O_4_ nanoneedle was synthesized via the template method following thermal annealing in air [156]. It exhibited a high Cs of 1440 C·cm^−2^ at 1 mA·cm^−2^ and a capacitance retention of 82.76% after 8000 cycles. Furthermore, the assembled Co_3_O_4_/MnCo_2_O_4_//AC SC device exhibited a high E of 31 Wh kg^−1^ at a P of 208.5 W·kg^−1^, and a capacitance retention of 101.23% after 8000 cycles.

#### 3.3.3. Co_3_O_4_/Metal Hydroxide Composites

Metal hydroxide mainly relies on a highly reversible redox reaction on the surface and in the body phase to store energy [157]. It is abundant and environmentally friendly and has an important application value in SCs [158]. For example, 3D nonstructural Co_3_O_4_/Co(OH)_2_ electrode material was synthesized using the hydrothermal method following electrodeposition (Figure 12a) [159]. It exhibited a high Cs of 1876 C·g^−1^ at 5 mA·cm^−2^ and a capacitance retention of 83.1% over 1000 cycles at 25 mA·cm^−2^. The performance depended on its structure. In the composite structure, Co(OH)_2_ nanosheets could grow both in space and on top of the Co_3_O_4_ nanotubes. In addition, a Co_3_O_4_/Ni(OH)_2_ nanosheet core–shell electrode material was directly grown on the surface of NF substrate [160]. Due to the unique properties of 2D nanomaterials, Co_3_O_4_/Ni(OH)_2_ electrode material effectively increased the active surface reaction with the electrolyte and facilitated ion diffusion. It displayed a high Cs of 1306.3 F·g^−1^ at 1.2 A·g^−1^, a capacitance retention of approximately 90% after 3000 cycles, and a capacity retention of about 46% at 12 A·g^−1^. The assembled Co_3_O_4_/Ni(OH)_2_//AC solid-state asymmetric SC exhibited an E of 40 Wh·kg^−1^ at a P of 3455 W·kg^−1^.

Through a large number of investigations, it was found that the performance of hydroxide electrode materials with a single metal element could not meet the actual needs in many aspects. [161] Therefore, two or more metal elements of hydroxides were often combined to achieve complementary advantages. For example, a core–shell nanosheet structural Co_3_O_4_/CoNi-layered double hydroxide (LDH) electrode material was deposited on NF via the growth of a nanoplate CoNi-LDH shell on the surface of the Co_3_O_4_ core (Figure 12b) [162]. It exhibited a high Cs of 2676.9 F·g^−1^ at 0.5 A·g^−1^ and a capacitance retention of 67.7% after 10,000 cycles at 30 A·g^−1^. The assembled asymmetric Co_3_O_4_/CoNi-LDH//AC device showed a high E of 61.23 Wh·kg^−1^ at a P of 750 W·kg^−1^. In addition, a family of layered double hydroxide nanosheet (IML; I = Ni, Co; M = Mn, Co, Al, L = LDH) coated Co_3_O_4_ nanowire arrays on NF was prepared via the hydrothermal method following the in situ growth method (Figure 12c) [163]. The LDH electrode material was expanded in interplanar spacing by the glucose molecule intercalation. Benefiting from the structure of Co_3_O_4_/IML, the Co_3_O_4_/NiMn-LDH electrode material exhibited an excellent Cs of 1644 F·g^−1^ at 1 A·g^−1^ and a cycling stability of 94.2% after 5000 cycles. The assembled Co_3_O_4_/NiMn-LDH//AC asymmetric SC device showed an E of 38.4 W h·kg^−1^ at a P of 800 W·kg^−1^. In addition to binary metal hydroxides, a core–shell structured Co_3_O_4_/NiCoAl-LDH hybrid electrode material was synthesized via the two-step hydrothermal method (Figure 12d) [164]. The Co_3_O_4_/NiCoAl-LDH nanowire electrode material exhibited a high Cs of 1104 F·g^−1^ at 1 A·g^−1^, a rate capability of 663 F·g^−1^ at 20 A·g^−1^, and a capacitance retention of 87.3% over 5000 cycles.

**Figure 12 molecules-25-00269-f012:**
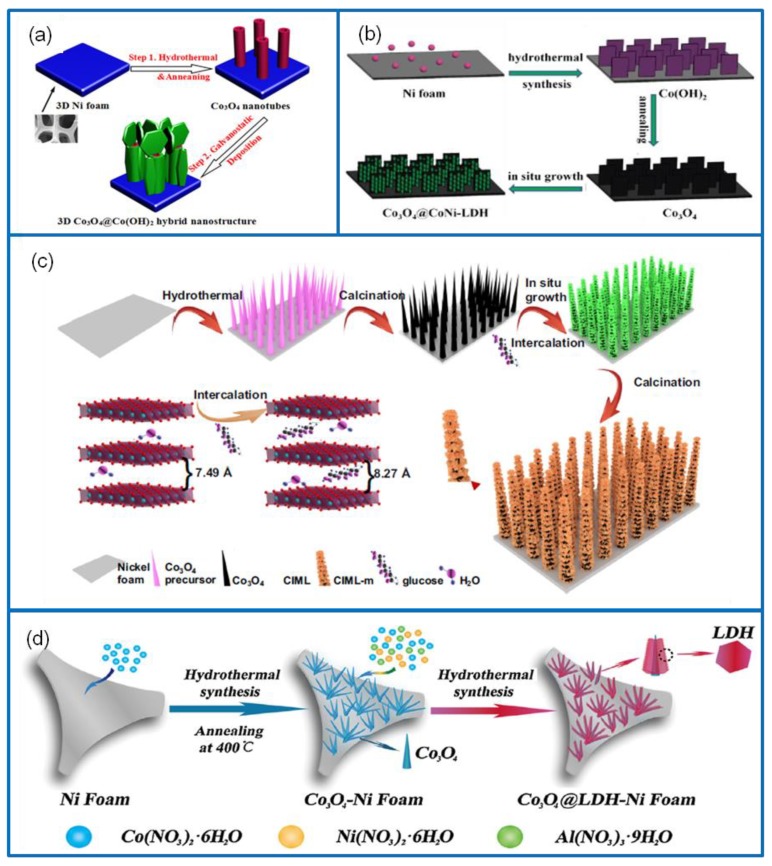
(**a**) Schematic route to synthesize 3D nonstructural Co_3_O_4_/Co(OH)_2_ electrode material [159]. (**b**) Schematic illustration of preparing Co_3_O_4_/CoNi-layered double hydroxide (LDH) nanowire electrode material [162]. (**c**) Schematic illustration of preparing the Co_3_O_4_/IML electrode materials and expanding interplanar spacing by the glucose molecule intercalation [163]. (**d**) Fabrication of the core–shell structural Co_3_O_4_/NiCoAl-LDH hybrid electrode material [164].

#### 3.3.4. Co_3_O_4_/Metal Sulfide Composites

Metal sulfide is considered to be a potentially high-performance electrode material, such as CoS, Co_3_S_4_, Ni_3_S_2_, CdS, Cu_2_S, and Ag_2_S [165]. Due to its low cost, high conductivity, mechanical property, and electrochemical activity, metal sulfide has become a kind of popular electrode material for Cs. Moreover, it has a rich redox reaction valence state, which helps with obtaining a high Cs. For the Co_3_O_4_ composite, 3D core–shell Co_3_O_4_/CoS nanosheet arrays were prepared on CC via the two-step electrodeposition method (Figure 13a) [166]. It exhibited a Cs of 764.2 F·g^−1^ at 1 A·g^−1^, a rate retention of 72.2% at 10 A·g^−1^, and a capacitance retention of 78.1% after 5000 cycles at 5 A·g^−1^. At the same time, Yan et al. [167] synthesized another Co_3_S_4_/Co_3_O_4_ core–shell structure with a different Co/S ratio via the hydrothermal lattice anion exchange method. The composite showed a Cs of 899 F·g^−1^ at 4 A·g^−1^ and a capacitance retention of 89.6% after 10,000 cycles. Further, a 3D core–shell of Co_3_O_4_/Ni_3_S_2_ nanowire electrode material was fabricated on NF via the hydrothermal method following electrodeposition [168]. It exhibited a high Cs of 1710 F·g^−1^ at 1 A·g^−1^, a rate capability capacitance of 86.2% at 10 A·g^−1^, and a capacitance retention of 85.3% after 1000 cycles. The assembled asymmetric Co_3_O_4_/Ni_3_S_2_//AC SC device showed a Cs of 126.6 F·g^−1^ at 1 A·g^−1^, a retention capacity of 88.5% over 5000 cycles, and a high E of 44.9 Wh·kg^−1^ at a P of 798 W·kg^−1^.

In addition, a core–shell Co_3_O_4_/CdS nanostructure was prepared on NF via the hydrothermal method following the ionic layer adsorption and reaction (ILAR) method (Figure 13b) [169]. It exhibited a high Cs of 1539 F·g^−1^ at 10 mV·s^−1^ and a capacitance retention of 98.5% after 2000 cycles. The symmetric Co_3_O_4_/CdS//Co_3_O_4_/CdS SC device displayed a Cs of 360 F·g^−1^ at 10 mV·s^−1^, a capacitance retention of 92% after 2000 cycles, and a high E of 40 Wh·kg^−1^ at 10 mA. Recently, 3D Cu_2_S and Ag_2_S metal sulfides combined Co_3_O_4_ nanostructure electrode materials were synthesized via the one-pot approach followed by the ILAR method (Figure 13c) [170]. The resulting 3D Co_3_O_4_/Cu_2_S and Co_3_O_4_/Ag_2_S electrode materials offered a highly conductive network, which achieved a quick transfer of ions and electrons and reduced internal resistance. The Co_3_O_4_/Cu_2_S electrode material exhibited an areal capacitance of 5324 mF·cm^−2^ at 10 mV·s^−1^, a good rate capability of 1630 mF·cm^−2^ at 100 mV·s^−1^, and a capacitance retention of 98.2% after 2000 cycles. Meanwhile, the Co_3_O_4_/Ag_2_S composite showed an areal capacitance of 2243 mF·cm^−2^ at 10 mV·s^−1^, a rate capability of 966 mF·cm^−2^ at 100 mV·s^−1^, and a capacitance retention of 96.7% after 2000 cycles. When they were assembled in a symmetrical SCs, the Co_3_O_4_/Cu_2_S//Co_3_O_4_/Cu_2_S electrode exhibited an areal capacitance of 1080 mF·cm^−2^ and a capacitance retention of 93.2% after 2000 cycles, while the Co_3_O_4_/Ag_2_S//Co_3_O_4_/Ag_2_S SC showed an areal capacitance of 645 mF·cm^−2^ and a capacitance retention of 92.8% over 2000 cycles.

### 3.4. Co_3_O_4_/Multiple Materials Composites

In addition to the above binary composites containing Co_3_O_4_, multiple-component materials containing Co_3_O_4_ have already been extensively reported [171,172,173,174,175]. Due to the synergistic effect of multiple components, the unique interfacial structure in multiple-component materials ensures an excellent electrochemical performance. A tremelliform Co_3_O_4_/NiO/Mn_2_O_3_ material was prepared via the template method [176]. It exhibited a high Cs of 3652 mF·cm^−2^ at 1 mA·cm^−2^, a rate property of 70% at 20 mA·cm^−2^, and a capacitance retention of 87.6% after 10,000 cycles. The performance was much better than that of Co_3_O_4_/NiO and Co_3_O_4_/Mn_2_O_3_. Moreover, asymmetrical Co_3_O_4_/NiO/Mn_2_O_3_//rGO SC showed a high E of 65.7 Wh·kg^−1^ at a P of 343.4 W·kg^−1^ within a voltage window of 1.7 V and a capacitance retention of 82% after 5000 cycles at 10 mA·cm^−2^. A hollow polyhedral network-like Co_3_O_4_/NiCo_2_O_4_/ZnCo_2_O_4_ electrode material was prepared via the coprecipitation and template method (Figure 14a) [177]. It exhibited a high Cs of 1892.5 F·g^−1^ at 1 A·g^−1^, a rate capacitance of 1135 F·g^−1^ at 10 A·g^−1^, and a capacitance retention of 66% over 2000 cycles. The assembled Co_3_O_4_/NiCo_2_O_4_/ZnCo_2_O_4_//AC SC device delivered a Cs of 233.75 F·g^−1^ at 1 A·g^−1^, a capacitance retention of 92% over 3000 cycles, and a high E of 83.11 Wh·kg^−1^ at a P of 800 W·kg^−1^. Additionally, a multielement Co_3_O_4_/C/MnO_2_ heterostructure on NF was prepared via a stepwise method (Figure 14b) [178]. It delivered a high Cs of 1561.3 F·g^−1^ at 0.5 A·g^−1^, a rate capability of 1335.3 F·g^−1^ at 20 A·g^−1^, and a capacitance retention of 95% over 10,000 cycles. Compared with its counterpart, the improved performance was attributed to the synergistic effect between multicomponents and C, which promoted the electron transfer between MnO_2_ and Co_3_O_4_.

Further, a high performance Co_3_O_4_/CNT/SS electrode material was fabricated via the electrodeposition method [179]. By adding SS core as a flow collector to CNT, the conductivity and electrodeposition efficiency of Co_3_O_4_ electrode were greatly improved. On the basis of these merits, the Co_3_O_4_/CNT/SS electrode material showed a volumetric capacitance of 82.94 F·cm^−3^ at 0.02 V·s^−1^ and an E of 1.31 mWh·cm^−3^ at a P of 294.80 mW·cm^−3^. Recently, a freestanding and flexible Co_3_O_4_-PPy-rGO electrode material was synthesized via hydrothermal, thermal reduction, and the electrochemical deposition method [180]. It exhibited a Cs of 532.8 F·g^−1^ at 5 mV·s^−1^ and a capacitance retention of 100% after 700 cycles. Due to the synergistic effect of Co_3_O_4_, PPy, and RGO, the electrochemical performance was much better than that of a single component. Among the electrode materials, Co_3_O_4_ nanoparticles and PPy were used to improve capacitance and reduce resistance. The rGO nanosheet could provide a large surface area. In addition, a multicomponent Co_3_O_4_/NiCo_2_O_4_/NiO/C&S composite was fabricated via the solvothermal and thermal decomposition method [181]. It showed an ultralong cycling retention of 94.2% after 20,000 cycles at 3 A·g^−1^. This excellent performance was attributed to the synergies among multicomponents and the highly stable structure formed by incompletely carbonized C and vulcanized S.

### 3.5. Performance Comparison of Co_3_O_4_-Containing Composites

The electrochemical performance of Co_3_O_4_-containing composites is summarized in Table 2. In general, compared with Co_3_O_4_, the performance of Co_3_O_4_-containing composites has been improved greatly, such as through high Cs, a long cycling stability, and a high rate capability. This performance is related to the morphology, stability, and Co_3_O_4_ nanoarrays on conductive substrates. Various Co_3_O_4_ structures have been reported, including nanosheet, nanowire, nanorod, and nanoparticle shapes. Moreover, a high surface area and more active sites of the Co_3_O_4_ composite are also required. For example, the Co_3_O_4_/PANI [113] electrode composite increased the contact between electrode and electrolyte for a high surface area, which is extremely helpful in improving the Cs. As for improving the rate performance and cycling stability of Co_3_O_4_ composites, the design and preparation of Co_3_O_4_ nanoarrays on conductive substrates is an important pathway. The core–shell Co_3_O_4_@MnO_2_ [135] nanoarrays are a good example. As a result, the contact area between the Co_3_O_4_ nanoarrays and the electrolyte increases. This is favorable for the infiltration of electrolytes with relatively low internal resistance. The Co_3_O_4_ nanoarray can be directly combined with conductive substrate, which facilitates the transfer of electrons, shortens the distance of electron/ion migration, and enhances mechanical stability. Therefore, low internal resistance and high electrochemical stability will lead to a high-rate performance and cycling stability. In addition, the synergistic effect of different components in composites is much more important. The Co_3_O_4_/C/MnO_2_ [178] composite delivered a better electrochemical performance. The electrical conductivity of Co_3_O_4_/C/MnO_2_ was improved through the introduction of C. The synergistic effect of the MnO_2_ nanosheet shell on lily-like nanostructures Co_3_O_4_ core could promote electron transfer and reduce the impedance and ion diffusion resistance of electrolytes. In addition, the Co_3_O_4_ nanostructure can be used as a better support and template to obtain composites with three-dimensional porous topology. 

## 4. Conclusions and Perspective

### 4.1. Conclusions

In summary, we have presented the recent progress on Co_3_O_4_ and Co_3_O_4_-containing electrode materials for high-performance SCs. Various synthetic methods and the electrochemical performance of Co_3_O_4_ electrode materials are summarized. The electrochemical performance of Co_3_O_4_ is compared according to the synthetic methods. Among these methods, solvothermal is the one most commonly used to synthesize Co_3_O_4_ electrode materials. Because the Co_3_O_4_ nanoparticles synthesized via the solvothermal method have a smaller size and a larger specific surface area, they offer more active sites. However, more and more researchers are paying attention to the template method. Compared to the solvothermal method, the template method can be used to design and synthesize Co_3_O_4_ nanomaterials. The products have the same morphology and size as the template. We can perfectly synthesize Co_3_O_4_ with a certain nanomorphology and pore structure, which increases the contact between electrode and electrolyt, and facilitates the charge transfer. For instance, Co_3_O_4_ electrode materials with ultrathin nanosheets often show a high electrochemical performance.

For Co_3_O_4_-containing electrode materials, we introduced the candidate hybrid materials, including carbon, conductive polymer, and metal compound materials. The multiple composites show better electrochemical performance due to the synergistic effect of several components, which effectively improve the Cs, E, P, cycling stability, and rate capacity of SCs. To date, Co_3_O_4_/graphene composites are the hottest research topic in SCs. Further, the composites consisting of Co_3_O_4_ and conductive polymers exhibited an excellent electrochemical performance, which is due to the easy diffusion of electrons and ions in conductive polymers. In addition, the abundant active sites in conductive polymers also facilitate the diffusion of electrolytes. In recent years, Co_3_O_4_/ternary metal oxide and Co_3_O_4_/metal sulfide composites have become the research focus. So far, SCs have been used in all aspects of society and have become an indispensable energy storage device. The wide application puts forward a high requirement for the specific capacitance and specific energy. Therefore, the design and development of Co_3_O_4_ and Co_3_O_4_-containing composites are urgently required to further improve the performance of SCs.

### 4.2. Perspective

Although great progress has been achieved in Co_3_O_4_ and Co_3_O_4_-containing electrode materials, there are still some problems to be solved in the application for SCs, such as the big gap between actual and theoretical specific capacitance, poor electrical conductivity, lower specific energy, and unclear development direction of Co_3_O_4_ and Co_3_O_4_-containing electrode materials. In order to solve these problems, some suitable strategies are put forward as follows:

(1) Designing Co_3_O_4_ and Co_3_O_4_-containing electrode materials with a high specific surface area and abundant porous structure. The large active surface of electrode materials is beneficial to electrolyte contact and adsorption of ions, which can decrease the electrolyte starvation near the electrode surface and facilitate ionic diffusion in the electrode. The unique porous structure ensures an efficient Faradaic reaction from outside to inside, leading to capacitance improvement. Therefore, a suitable synthesis method is urgently needed to control the specific morphology and structure of Co_3_O_4_ and Co_3_O_4_-containing electrode materials.

(2) Hybridizing Co_3_O_4_ with conductive materials, including PANI, PPy, CNT, graphene, etc. The candidate materials should be considered to enhance the conductive feature of Co_3_O_4_, so as to improve the rate performance and cycling stability of Co_3_O_4_-containing composites. Meanwhile, the ordered Co_3_O_4_ nanoarrays provide a large surface area and short diffusion path for ion transfer, which will contribute much more pseudocapacitance.

(3) Developing novel Co_3_O_4_ and Co_3_O_4_-containing electrode materials. Novel structures and morphologies depend on the synthesis methods and synthesis conditions. Therefore, the development of various Co_3_O_4_ microstructures is still an important task. In terms of hybridization, we should continuously explore suitable substances and make use of the synergistic effects between multiple substances to further improve the electrochemical performance of Co_3_O_4_.

(4) Enlarging the working voltage window of the Co_3_O_4_ and Co_3_O_4_-containing electrode materials. In addition to increasing capacitance, enlarging operating voltage is also an effective way to increase specific energy. Due to the easy decomposition and narrow voltage window of aqueous electrolytes, organic electrolytes, ionic liquids, and solid-state/gel electrolytes should be considered to expand the working voltage window, so as to improve the specific energy of Co_3_O_4_ and Co_3_O_4_-containing composites.

## Figures and Tables

**Figure 1 molecules-25-00269-f001:**
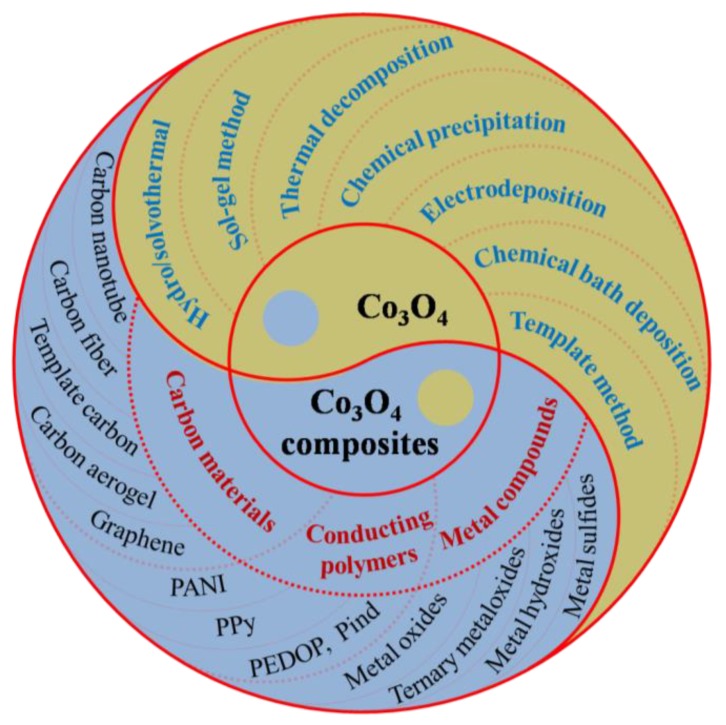
Schematic illustration of Co_3_O_4_ and Co_3_O_4_-containing composites materials for supercapacitors (SCs).

**Figure 3 molecules-25-00269-f003:**
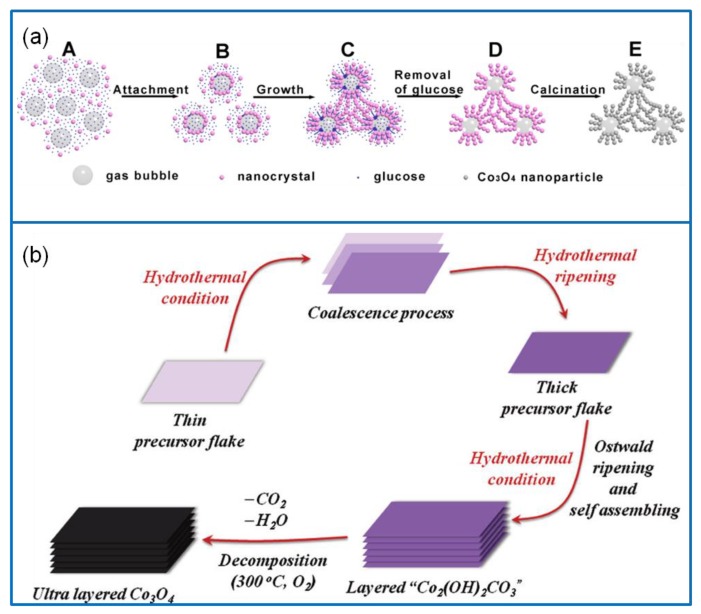
(**a**) Schematic illustration of preparing hollow structure Co_3_O_4_ via the direct precipitation method [54]. (**b**) Schematic illustration of preparing ultralayered Co_3_O_4_ via the homogeneous precipitation method [56].

**Figure 4 molecules-25-00269-f004:**
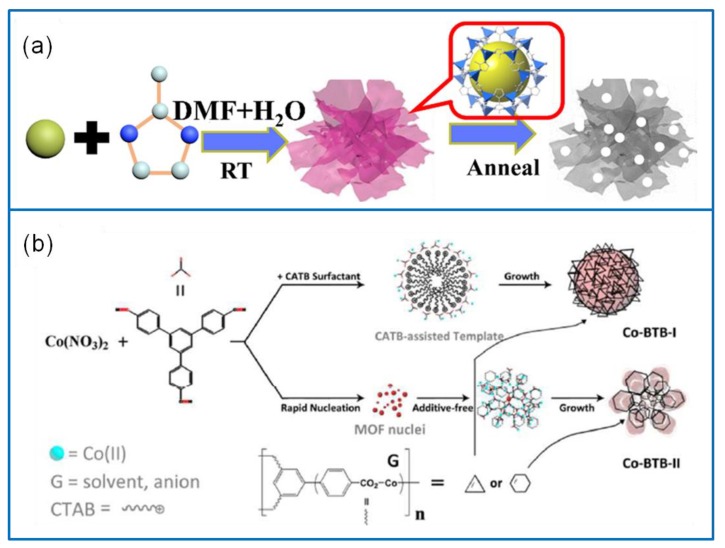
(**a**) Synthesis process of ultrathin Co_3_O_4_ via the template method [67]. (**b**) Schematic illustration of hollow spherical Co-BTB-I and the flower-like Co-BTB-II [68].

**Figure 5 molecules-25-00269-f005:**
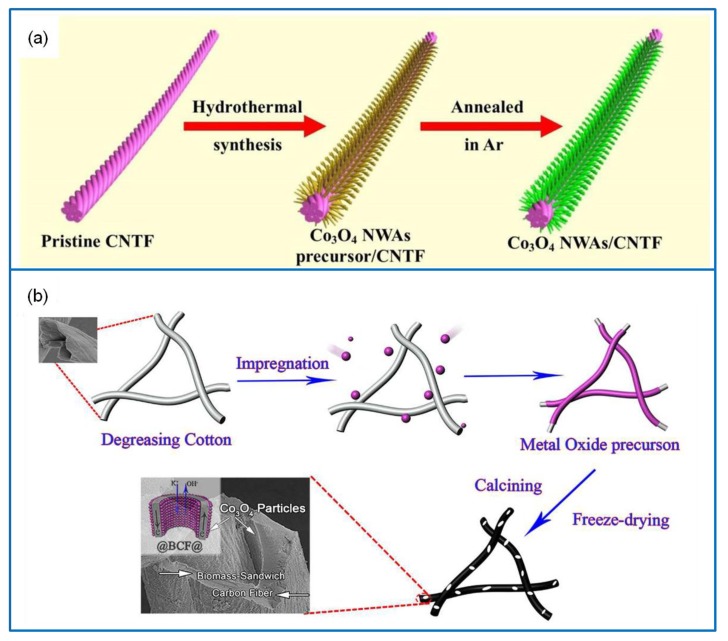
(**a**) Schematic illustration of synthesizing the 3D Co_3_O_4_ nanowire arrays directly on carbon fibers (CFs) [84]. (**b**) Fabrication of hierarchical Co_3_O_4_/biomass-derived CF (BCF) electrode material [85].

**Figure 6 molecules-25-00269-f006:**
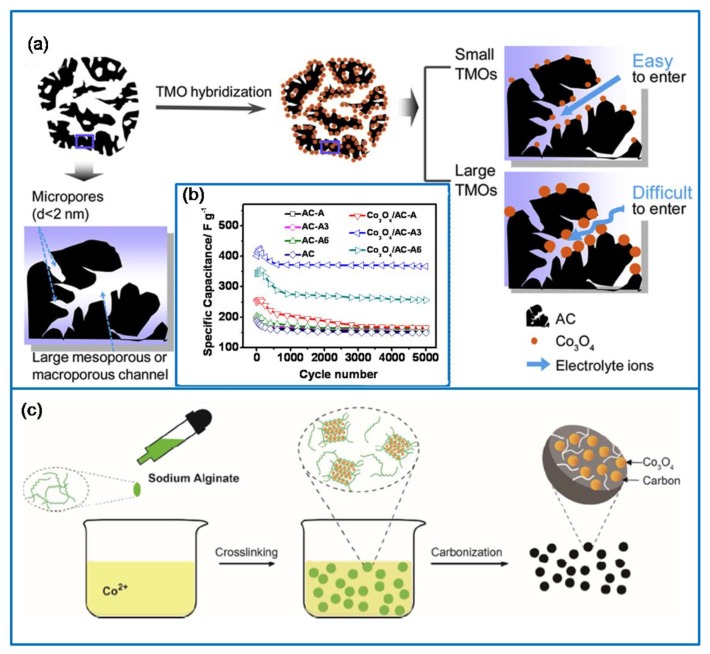
(**a**) Schematic of controlling the Co_3_O_4_ particles size. (**b**) The cycling life performance of AC- and Co_3_O_4_/AC-based electrodes at 5 A·g^−1^ [91]. (**c**) Synthesis process of the Co_3_O_4_/AC electrode materials [92].

**Figure 7 molecules-25-00269-f007:**
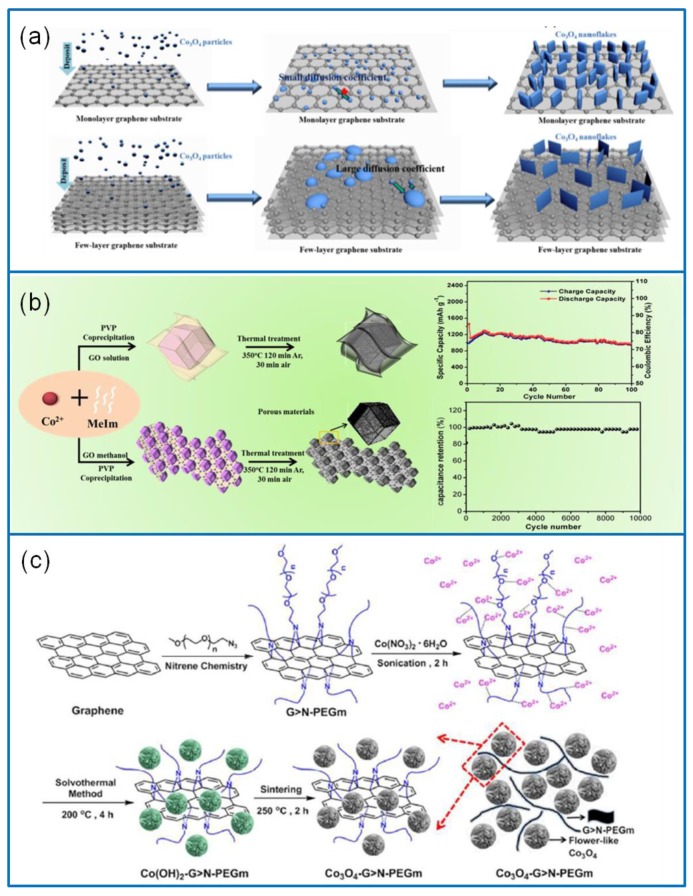
(**a**) Schematic illustration of synthesizing Co_3_O_4_/graphene@Ni electrode materials via the in situ synthesis method [106]. (**b**) Schematic route to prepare the Co_3_O_4_/rGO electrode composites [107]. (**c**) Modifying process of Co_3_O_4_-G > N electrode material by mPEG [108].

**Figure 8 molecules-25-00269-f008:**
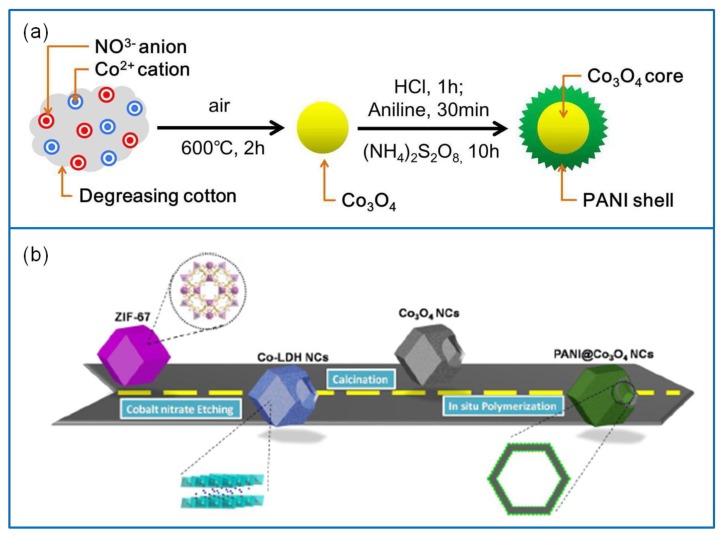
(**a**) Preparation of core–shell structured Co_3_O_4_/ polyaniline (PANI) electrode material via the in situ polymerization method [113]. (**b**) Schematic illustration of synthesizing hierarchically hollow Co_3_O_4_/PANI electrode material via the in situ polymerization route [114].

**Figure 9 molecules-25-00269-f009:**
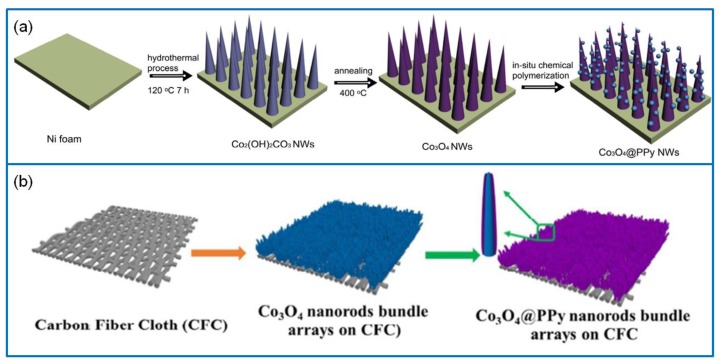
(**a**) Schematic illustration of preparing the hierarchical Co_3_O_4_/polypyrrole (PPy) core–shell composite nanowires [118]. (**b**) Fabrication of a flexible, high-performance, and tailorable nanorod Co_3_O_4_/PPy electrode [120].

**Figure 10 molecules-25-00269-f010:**
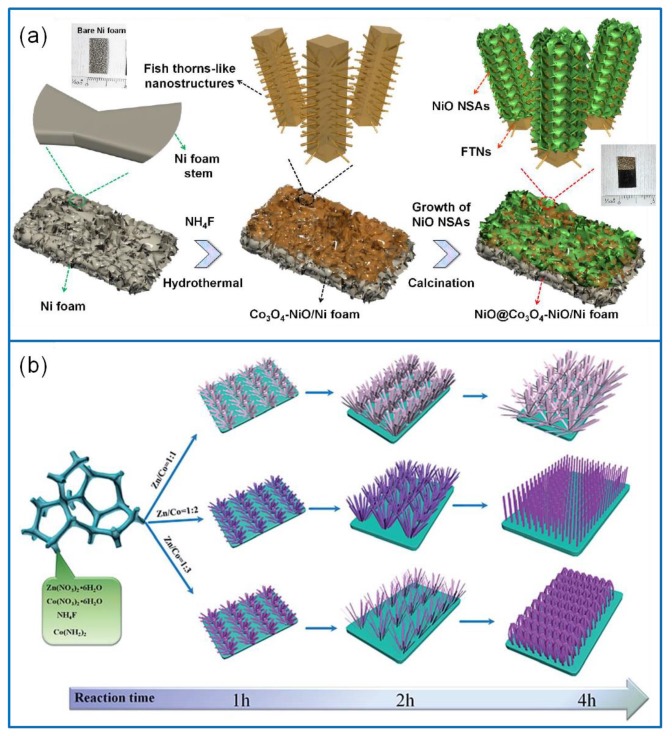
(**a**) Schematic illustration of forming NiO/Co_3_O_4_/NiO core–shell-like electrode material [131]. (**b**) Schematic route to synthesize hierarchical flower-like Co_3_O_4_/ZnO nanobundles [132].

**Figure 11 molecules-25-00269-f011:**
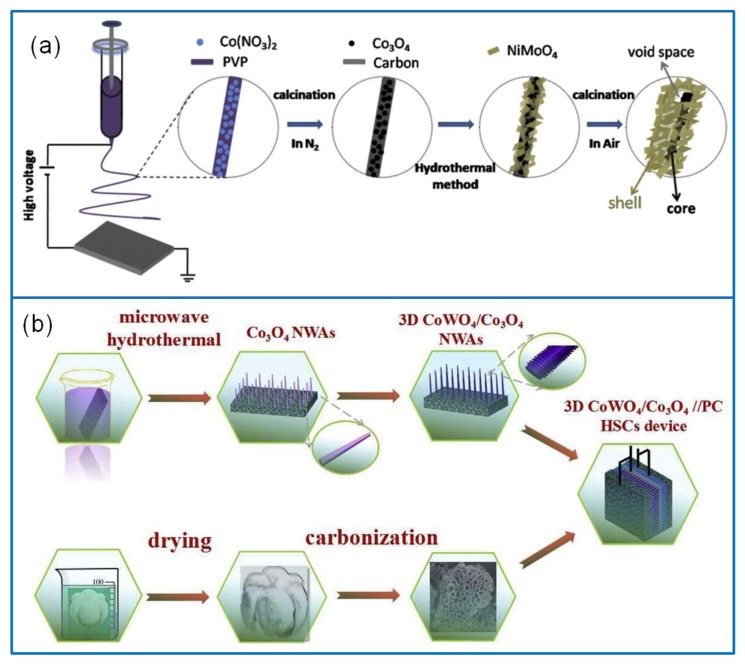
(**a**) Schematic illustration of preparing the Co_3_O_4_/NiMoO_4_ electrode materials [151]. (**b**) Fabrication of 3D heterostructure CoWO_4_/Co_3_O_4_//porous carbon SC device via the microwave hydrothermal method [152].

**Figure 13 molecules-25-00269-f013:**
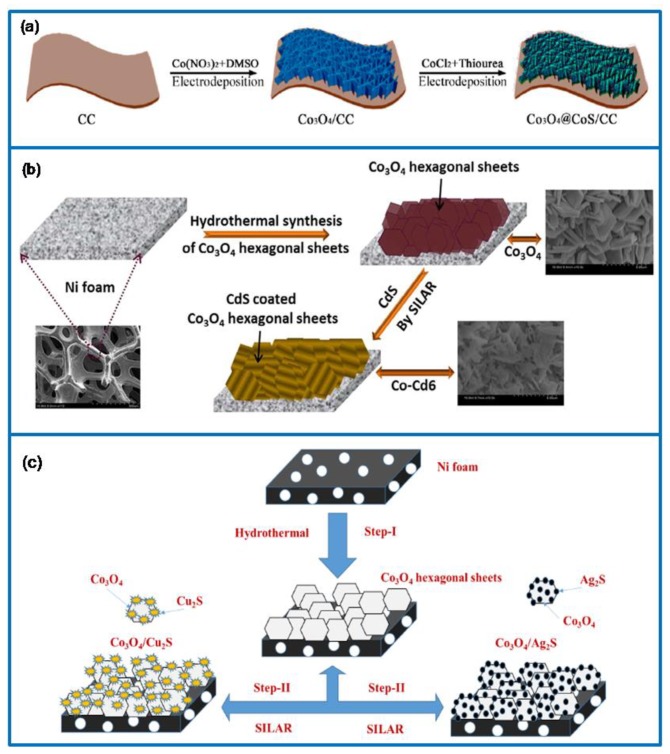
(**a**) Synthesis process of Co_3_O_4_/CoS nanosheet arrays on CC [166]. (**b**) Schematic illustration of preparing Co_3_O_4_/CdS on nickel foam (NF) [169]. (**c**) Schematic route to synthesize 3D Co_3_O_4_/Cu_2_S and Co_3_O_4_/Ag_2_S electrode materials [170].

**Figure 14 molecules-25-00269-f014:**
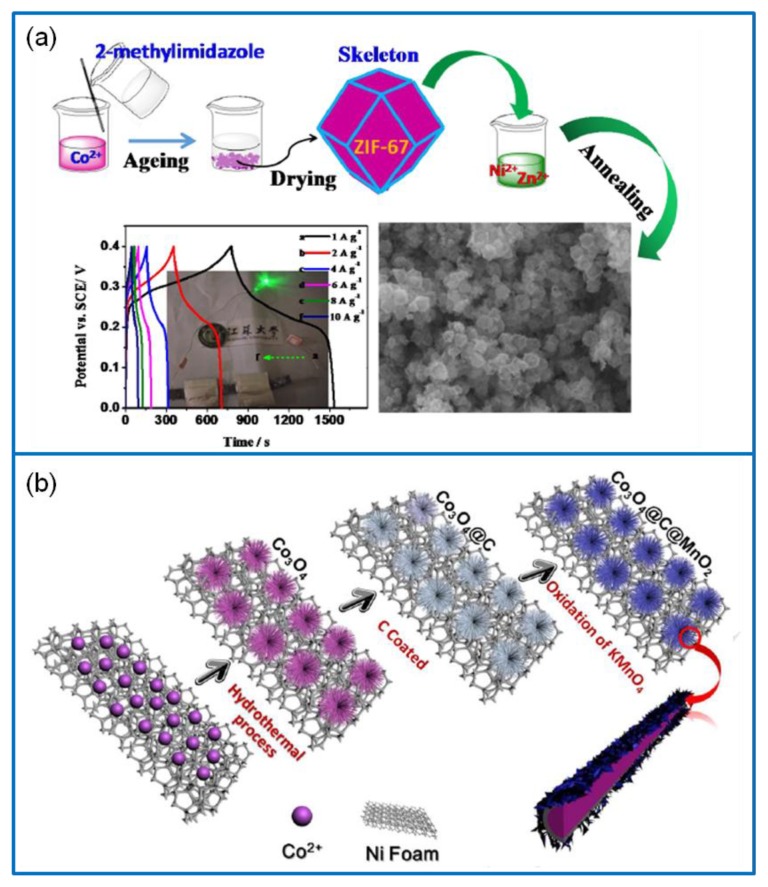
(**a**) Schematic route to synthesize the hollow polyhedral network-like Co_3_O_4_/NiCo_2_O_4_/ZnCo_2_O_4_ composite [177]. (**b**) Fabrication of the heterostructural Co_3_O_4_/C/MnO_2_ electrode material [178].

**Table 1 molecules-25-00269-t001:** Electrochemical performance of Co_3_O_4_ electrode materials.

Synthetic Method	Material Structure	Specific Capacitance	Rate	Cycle Life Retention	Ref.
Hydrothermal	Nanosheet	610 F·g^−1^ at 1 A·g^−1^	65.7% at 10 A·g^−1^	94.5% after 3000 cycles	[37]
Hydrothermal	Nanoflake	1500 F·g^−1^ at 1 A·g^−1^	55.2% at 10 A·g^−1^	99.3% after 2000 cycles	[39]
Solvothermal	Nanoparticle	523.0 F·g^−1^ at 0.5 A·g^−1^	66.9% at 5 A·g^−1^	104.9% after 1500 cycles.	[42]
Solvothermal	Nanosphere	837.7 F·g^−1^ at 1 A·g^−1^	93.6% at 10 A·g^−1^	87.0% after 2000 cycles	[43]
Sol–gel	Nanoparticle	120 F·g^−1^ at 1 A·g^−1^	_	_	[46]
Sol–gel	Netlike	708 F·g^−1^ at 5 mV·s^−1^	71.9% at 50 mV·s^−1^	_	[48]
Thermal decomposition	Nanowire	2815.7 F·g^−1^ at 1 A·g^−1^	27.2% at 20 A·g^−1^	88.8% after 1100 cycles	[50]
Thermal decomposition	Nanoflake	576.8 F·g^−1^ at 1 A·g^−1^	49.2% at 50 A·g^−1^	82% after 5000 cycles.	[51]
Chemical precipitation	Nanonet	739 F·g^−1^ at 1 A·g^−1^	72.1% at 15 A·g^−1^	90.2% after 1000 cycles	[54]
Chemical precipitation	Ultralayer	548 F·g^−1^ at 8 A·g^−1^	66% at 32 A·g^−1^	98.5% after 2000 cycles	[56]
Electrodeposition	Nanoplate	517 F·g^−1^ at 1 A·g^−1^	39.1% at 20 A·g^−1^	91% after 3000 cycles.	[58]
Electrodeposition	Nanosheet	6469 F·g^−1^ at 5 mA·cm^−2^	63.8% at 15 mA·cm^−2^	81.6% after 2000 cycles	[60]
Chemical bath deposition	Nanowire	850 F·g^−1^ at 5 mV·s^−1^	~85 at 100 mV·s^−1^	86% after 5000 cycles.	[62]
Chemical bath deposition	Nanorod	387.3 F·g^−1^ at 1 A·g^−1^	33.6% at 5 A·g^−1^	88% after 1000 cycles.	[63]
Template	Ultrathin Nanosheet	1121 F·g^−1^ at 1 A·g^−1^	77.9% at 25 A·g^−1^	98.2% after 6000 cycles	[66]
Template	Ultrathin Nanosheet	1216.4 F·g^−1^ at 1 A·g^−1^	76.1% at 20 A·g^−1^	86.4% after 8000 cycles	[67]
Spray pyrolysis	Thin film	412 F·g^−1^ at 1 A·g^−1^	93% at 4 A·g^−1^	92.6% after 1000 cycles	[69]
Chemical vapor deposition	Nanosphere	128 F·g^−1^ at 10 mV·s^−1^	~90% at 20 A·g^−1^	>100% after 4000 cycles	[70]
Electrospinning technique	Nanofiber	340 F·g^−1^ at 1 A·g^−1^	87.1% at 10 A·g^-1^	94% after 1000 cycles	[71]
Galvanic displacement	Ultrathin nanosheet	1095 F·g^−1^ at 1 A·g^−1^	61.9% at 15 A·g^−1^	71% after 2000 cycles	[72]
Laser ablation	Nanosheet	762 F·g^−1^ at 6 A·g^−1^	82.7% at 36 A·g^−1^	-	[73]
In-site self-organization	Nanorods	1486 F·g^−1^ at 1 A·g^−1^	72.9% at 15 A·g^−1^	98.8% after 5000 cycles	[74]

**Table 2 molecules-25-00269-t002:** Electrochemical performance of Co_3_O_4_-containing electrode materials.

Materials	Co_3_O_4_ Structure	Specific Capacitance	Rate	Cycle life Retention	Ref.
Co_3_O_4_/SWCNT	Porous nanoflake	313.9 F·g^−1^ at 1 mV·s^−1^	39.6% at 20 mV·s^−1^	80% after 3000 cycles	[78]
Co_3_O_4_/MWCNT	Nanofiber	406 F·g^−1^ at 2 A·g^−1^	41.9% at10 A·g^−1^	93% after 10,000 cycles	[80]
Co_3_O_4_/CF	Nanoparticle	586 F·g^−1^ at 1 A·g^−1^	66% at 50 A·g^−1^	74% after 2000 cycles	[83]
Co_3_O_4_/CF	Nanoparticle	948.9 F·g^−1^ at 0.5 A·g^−1^	48.2% at 40 A·g^−1^	88% after 6000 cycles	[85]
Co_3_O_4_/AC	Nanoparticle	491 F·g^−1^ at 0.1 A·g^−1^	82% at 5 A·g^−1^	89% after 5000 cycles	[91]
Co_3_O_4_/TC	Nanoparticle	885 F·g^−1^ at 2.5 A·g^−1^	23.7% at 20 A·g^−1^	94% over 10,000 cycles	[96]
Co_3_O_4_/CA	Ultrafine nanoparticle	616 F·g^−1^ at 1 A·g^−1^	72.2% at 20 A·g^−1^	93.6% after 5000 cycles	[100]
Co_3_O_4_/CA	Nanowire	1167.6 F·g^−1^ at 1 A·g^−1^	42.8% at 50 A·g^−1^	92.4% after 10,000 cycles	[101]
Co_3_O_4_/grapheme	Nanofiber	1935 F·g^−1^ at 1 A·g^−1^	72.9% at 50 A·g^−1^	83% after 2000 cycles	[34]
Co_3_O_4_/grapheme	Flower-like microsphere	1625.6 F·g^−1^ at 0.5 A·g^−1^	-	87% after 5000 cycles	[108]
Co_3_O_4_/PANI	Spherical nanoparticle	1184 F·g^−1^ at 1.25 A·g^−1^	42.2% at 50 A·g^−1^	84.9% after 1000 cycles	[113]
Co_3_O_4_/PANI	Nanocage particle	1301 F·g^−1^ at 1 A·g^−1^	62.6% at 10 A·g^−1^	90% after 2000 cycles	[114]
Co_3_O_4_/PPy	Nanowire	2122 F·g^−1^ at 5 mA·cm^−2^	53.3% at 50 mA·cm^−2^	77.8% after 5000 cycles	[118]
Co_3_O_4_/PPy	Nanorod	6.67 F·cm^−2^ at 2 mA·cm^−2^	97.4% at 20 mA·cm^−2^	~100% after 2000 cycles	[120]
Co_3_O_4_/PEDOP	Nanorod	582 F·g^−1^ at 0.5 A·g^−1^	69.9% at 1 A·g^−1^	78% after 5000 cycles	[122]
Co_3_O_4_/Pind	Nanoparticle	1805 F·g^−1^ at 2 A·g^−1^	90% at 25 A·g^−1^	85% after 1000 cycles	[123]
Co_3_O_4_/NiO	Nanorod	313.9 μAh·cm^−2^ at 4 mA·cm^−2^	76.38% at 25 mA·cm^−2^	135% after 2000 cycles	[131]
Co_3_O_4_/ZnO	Flower-like nanobundle	1983 F·g^−1^ at 2 A·g^−1^	42% at 20 A·g^−1^	84.5% after 5000 cycles	[132]
Co_3_O_4_/CuO	Nanowire	1242 F·g^−1^ at 2 mV·s^−1^	51% at 50 mV·s^−1^	100% after 2000 cycles	[133]
Co_3_O_4_/CoO	Nanomicrosphere	3377.8 F·g^−1^ at 2 A·g^−1^	66.5% at 20 A·g^−1^	39.4% after 4000 cycles	[134]
Co_3_O_4_/MnO_2_	Nanowire	1920 F·g^−1^ at 1 A·g^−1^	-	95.2% after 3000 cycles	[135]
Mn-doping Co_3_O_4_	Nanoneedle	668.4 F·g^−1^ at 1 A·g^-1^	61.7% at 10 A·g^−1^	104% after 10,000 cycles	[155]
Fe-doping Co_3_O_4_	Flowerlike nanoflake	1997 F·g^−1^ at 1 A·g^−1^	61.7% at 20 A·g^−1^	92.1% after 5000 cycles	[138]
Co_3_O_4_/NiCo_2_O_4_	Nanosheet	1330 F·g^−1^ at 3 mA·cm^−2^	72.2% at 30 mA·cm^−2^	100.7% over 5000 cycles	[150]
Co_3_O_4_/NiMoO_4_	Nanofiber	998.05 F·g^−1^ at 0.5 A·g^−1^	88% at 20 A·g^−1^	89.9% after 3000 cycles	[151]
Co_3_O_4_/CoWO_4_	Nanocone	4.665 F·cm^−2^ at 2.7 mA·cm^−2^	70.83% at 27 mA·cm^−2^	-	[152]
Co_3_O_4_/ZnFe_2_O_4_	Nanocage particle	326.7 F·g^−1^ at 1 A·g^−1^	51.8% at 10 A·g^−1^	80.7% after 1000 cycles	[153]
Co_3_O_4_/CoFe_2_O_4_	Nanoparticle	761.1 F·g^−1^ at 10 mV·s^−1^	20.9% at 50 mV·s^−1^	92.2% after 1000 cycles	[154]
Co_3_O_4_/CoMn_2_O_4_	Nanosheet	1627 F·g^−1^ at 1 A·g^−1^	84.6% at 10 A·g^−1^	87.6% over 3000 cycles	[155]
Co_3_O_4_/MnCo_2_O_4_	Polyhedron nanoparticle	1440 C·cm^−2^ at 1 mA·cm^−2^	36% at 10 mA·cm^−2^	82.76% after 8000 cycles	[156]
Co_3_O_4_/Co(OH)_2_	Nanotube	1876 C·g^−1^ at 5 mA·cm^−2^	25.4% at 25 mA·cm^−2^	83.1% over 1000 cycles	[159]
Co_3_O_4_/Ni(OH)_2_	Nanosheet	1306.3 F·g^−1^ at 1.2 A·g^−1^	46% at 12 A·g^−1^	90% after 3000 cycles	[160]
Co_3_O_4_/CoNi-LDH	Nanoplate	2676.9 F·g^−1^ at 0.5 A·g^−1^	43% at 20 A·g^−1^	67.7% after 10,000 cycles	[162]
Co_3_O_4_/NiMn-LDH	Nanowire	1644 F·g^−1^ at 1A·g^−1^	42.4% at 10A·g^−1^	94.2% after 5000 cycles	[163]
Co_3_O_4_/CoS	Nanosheet	764.2 F·g^−1^ at 1.0 A·g^−1^	72.2% at 10 A·g^−1^	78.1% after 5000 cycles	[166]
Co_3_O_4_/Ni_3_S_2_	Nanowire	1710 F·g^−1^ at 1A·g^−1^	86.2% at 10 A·g^−1^	85.3% after 1000 cycles	[168]
Co_3_O_4_/CdS	Nanosheet	1539 F·g^−1^ at 10mV·s^−1^	52% at 100 mV·s^−1^	98.5% after 2000 cycles	[169]
Co_3_O_4_/Cu_2_S	Nanosheet	5324 mF·cm^−2^ at 10 mV·s^−1^	30.6% at 100 mV·s^−1^	98.2% after 2000 cycles	[170]
Co_3_O_4_/Ag_2_S	Nanosheet	2243 mF·cm^−2^ at 10 mV·s^−1^	43.1% at 100 mV·s^−1^	96.7% after 2000 cycles	[170]
Co_3_O_4_/NiO/Mn_2_O_3_	Nanosheet	3652 mF·cm^−2^ at 1 mA·cm^−2^	70% at 20 mA·cm^−2^	87.6% after 10,000 cycles	[176]
Co_3_O_4_/NiCo_2_O_4_/ZnCo_2_O_4_	Nanocage particle	1892.5 F·g^−1^ at 1 A·g^−1^	60% at 10 A·g^−1^	66% after 2000 cycles	[177]
Co_3_O_4_/C/MnO_2_	Lily-like nanostructures	1561.3 F·g^−1^ at 0.5 A·g^-1^	85.5% at 20 A·g^−1^	95% after 10,000 cycles	[178]
Co_3_O_4_/CNT/SS	Nanoparticle	82.94 F·cm^−3^ at 0.02 V·s^−1^	58.96 at 0.05 V·s^−1^	80.4 after 1000 cycles	[179]
Co_3_O_4_-PPy-rGO	Nanoparticle	532.8 F·g^−1^ at 5 mV·s^−1^	-	100% after 700 cycles	[180]
Co_3_O_4_/NiCo_2_O_4_/NiO/C&S	Nanoparticle	428.24 F·g^−1^ at 0.5 A·g^−1^	61.5% at 10 A·g^−1^	94.2% after 20,000 cycles	[181]

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
