# Peer review of "Recent Advance in Co3O4 and Co3O4-Containing Electrode Materials for High-Performance Supercapacitors"

_molecules, 2020, doi:10.3390/molecules25020269_

Round 1
Reviewer 1 Report
The authors present an useful review facing the features and challenges of Co3O4 based electrodes. Useful information and data is provided, which is obtained from suitable literature. Revision is recommended, among ohers, to obtain consistency with recent literature..
Introduction
“The energy stored and released amount is related to the speed in energy storage devices. In general, the greater the discharge power, the lower the energy that can be released.”
I think common literature would help for this statement:
The comparison of the possible electrochemical energy storage devices: Winter, R.J. Brodd, Chemical Reviews, 104 (2004) 4245-4269.2) The mathematical definition and relation of energy vs power:
Kasnatscheew, R. Wagner, M. Winter, I. Cekic-Laskovic, Topics in Current Chemistry, 376 (2018) 16.
“The performance of SCs mainly involves specific capacitance (Cs), energy density (E), power density (P), resistance, cycling stability, and rate capacity. They are obtained by electrochemical cyclic voltammetry (CV), galvanostatic charge/discharge (GCD), and electrochemical impedance spectroscopy (EIS) [2].”
“Specifc” implies gravimetric relation (per mass) while “density” implies volumetric relation (per volume). Please be consistent.
Conclusion
“Because the Co3O4 nanoparticles synthesized by solvothermal method have a smaller size and a larger specific surface area, which offer more active sites.”
Sounds incorrect: Either delete “because” or substitute “which” with “they”.
“However, more and more researchers pay attention to template method. It can perfectly synthesize Co3O4 with a certain nano morphology and pore structure, which increase the contact between electrode and electrolyte, and facilitates the charge transfer. For instance, the Co3O4 electrode materials with ultrathin nanosheets often show a high electrochemical performance.”
“Solvothermal method” vs “template method”: What is the disadvantage/challenge of template method?
“voltage window” vs “potential window”..please be more consistent
Reviewer 2 Report
Manuscript Number: molecules-667132 by Wang et al. titled “Recent Advance on Co3O4 and Co3O4-Containing Electrode Materials for High Performance Supercapacitors”
This is a manuscript on 171 references.
Wang et al. reviewed the advances in the synthesis and fabrication of Co3O4 and Co3O4 nanocomposite materials for applications in supercapacitors. This is a very important and rising topic in the research field of energy materials.
This manuscript can be published in the Molecules after a major revision. Below are the reviewer’s comments.
1) Some important reviews and recent advances in the use of Co3O4/graphene materials should be included in the revised version. The authors should discuss in the introduction the use of graphene and GO nanocomposites for supercapacitors. The authors need to briefly discuss in the introduction recent use of 2D nanomaterials in supercaapcitors[see / DOI: 10.1002/aenm.201502159 /] and https://doi.org/10.1016/j.electacta.2016.02.012] Graphene composite supercapacitor: Adv. Energy Mater. 2015, 5, 1401890; Asymmetric supercapacitor: Adv Funct Mater 2015, 25, 7291) and Adv. Mater. 2015, 27, 6714. These references should be included in the revised manuscript.
2) A discussion on the use of Co3O4 nanocomposites in Asymmetric supercapacitors should be included in the revised manuscript.
3) The stability of Co3O4 composites with pseudocapacitor should be discussed.
4) It is important to note that except for MnO2 and RuO2, other ternary transition metal oxides (based on cobalt) are also used as potential supercapacitor materials with good performance, such as NiCo2O4, ZnCo2O4.( Dong, B., Zhang, X., Xu, X., Gao, G., Ding, S., Li, J., & Li, B. Preparation of scale-like nickel cobaltite nanosheets assembled on nitrogen-doped reduced graphene oxide for high-performance supercapacitors. Carbon, 2014, 80, 222-228. Wang, W., Yang, Y., Yang, S., Guo, Z., Feng, C., & Tang, X.
5) This manuscript contains a few technically flawed claims and English errors and must be significantly revised and proofread by English language expert.
Round 2
Reviewer 2 Report
The authors revised their manuscript carefully and responded to all the comments and questions raised by this reviewer. Therefore, the revised manuscript is suitable for publication in Molecules